# Hydroxylated $TiO_2$-induced high-density Ni clusters for breaking the activity-selectivity trade-off of $CO_2$ hydrogenation

Cong-Xiao Wang[1,6], Hao-Xin Liu[1,6], Hao Gu[2,6], Jin-Ying Li[1], Xiao-Meng Lai[1], Xin-Pu Fu[1], Wei-Wei Wang [1], Qiang Fu [3,4], Feng Ryan Wang [2] ✉, Chao Ma [5] ✉ & Chun-Jiang Jia [1] ✉

The reverse water gas shift reaction can be considered as a promising route to mitigate global warming by converting $CO_2$ into syngas in a large scale, while it is still challenging for non-Cu-based catalysts to break the trade-off between activity and selectivity. Here, the relatively high loading of Ni species is highly dispersed on hydroxylated $TiO_2$ through the strong Ni and −OH interactions, thereby inducing the formation of rich and stable Ni clusters (~1 nm) on anatase $TiO_2$ during the reverse water gas shift reaction. This Ni cluster/$TiO_2$ catalyst shows a simultaneous high $CO_2$ conversion and high CO selectivity. Comprehensive characterizations and theoretical calculations demonstrate Ni cluster/$TiO_2$ interfacial sites with strong $CO_2$ activation capacity and weak CO adsorption are responsible for its unique catalytic performances. This work disentangles the activity-selectivity trade-off of the reverse water gas shift reaction, and emphasizes the importance of metal−OH interactions on surface.

As an available but inert source of carbon, greenhouse gas $CO_2$ can be converted by hydrogen to form CO, which can be used as the feedstock for synthetic fuels[1–6]. Except for Cu-based catalysts, the selective hydrogenation of $CO_2$ to CO, known as the reverse water gas shift (RWGS) reaction, is always accompanied by severe methanation, especially for the low-cost Ni-based catalysts with high C−O bond cleavage capacity[7–9]. Currently, methanation can be effectively inhibited through reported methods, including reducing metal loading[10–12], forming metal carbide/phosphide[2,13], applying the encapsulation layer over metal sites[14–16], decreasing particle size[9], and varying the support materials[17]. However, these methods decrease the density of surface active sites, leading to a significant activity trade-off for high CO selectivity. Therefore, it is important but challenging to develop a strategy to break such activity-selectivity trade-off over non-Cu-based catalysts[18–20].

To enable a simultaneous high CO selectivity at high $CO_2$ conversion, a catalyst must satisfy two key requirements[21]. On the one hand, sufficient active sites are required to activate and dissociate $CO_2$[22–24]. On the other hand, active sites must possess relatively weak CO adsorption capacity, thus preventing the C−H bond formation and further hydrogenation to $CH_4$[2,25]. It has been reported that compared to metal nanoparticles, supported metal clusters (SMCs) in high oxidation states have less adsorbing strength to CO, allowing a higher CO selectivity in $CO_2$ hydrogenation[22,26]. However, the stability of SMCs is usually negatively correlated with their surface density under harsh $CO_2$ hydrogenation conditions, leading to cluster sintering and rapid deactivation[9,27,28].

As one typical reducible support, $TiO_2$ with active surface oxygen has been widely used in supporting active metals to catalyze RWGS reaction[29,30]. Unfortunately, due to its limited ability to disperse metal

[1]Key Laboratory for Colloid and Interface Chemistry, Key Laboratory of Special Aggregated Materials, School of Chemistry and Chemical Engineering, Shandong University, Jinan 250100, China. [2]Department of Chemical Engineering, University College London, Roberts Building, Torrington Place, London WC1E 7JE, UK. [3]Hefei National Research Center for Physical Sciences at the Microscale, University of Science and Technology of China, Hefei 230026, China. [4]School of Future Technology, University of Science and Technology of China, Hefei, 230026, China. [5]College of Materials Science and Engineering, Hunan University, Changsha 410082, China. [6]These authors contributed equally: Cong-Xiao Wang, Hao-Xin Liu, Hao Gu. ✉e-mail: ryan.wang@ucl.ac.uk; cma@hnu.edu.cn; jiacj@sdu.edu.cn

species, metal loadings above 2 wt% usually trigger the aggregation of SMCs on $TiO_2$ and, with selectivity, switch to methanation[31,32]. A feasible pathway to overcome the above challenges is to construct anchoring sites on the $TiO_2$ surface that increase the binding strength to SMCs during the reaction. Recent reports suggest that surface hydroxyl (−OH) can be considered an effective anchoring site for dispersing active metals[33–36]. Even at room temperature, enriched surface −OH can effectively redisperse sintered metal particles into single atoms and clusters. Therefore, it can be expected that the sinter-resistant cluster catalyst can be designed by modifying the $TiO_2$ surface with sufficient −OH, a method yet to be reported.

Herein, by converting the commercial anatase $TiO_2$ into $H_2Ti_3O_7$ tubes with abundant surface −OH groups, Ni species with relatively high loading (10 wt%) are dispersed as isolated atoms anchoring via the strong −OH·Ni binding force. Under the reducing atmosphere, accompanied by the removal of surface −OH and the reduction of Ni species, isolated Ni atoms are converted into stable clusters and a few $TiO_x$-covered particles. In contrast, Ni species aggregate into large-size particles with poor activity on the reference commercial $TiO_2$. Comprehensive characterizations and theoretical simulations confirm that due to the presence of Ni cluster/$TiO_2$ interfaces with improved $CO_2$ activation and weak CO adsorption, the activity-selectivity trade-off of the RWGS reaction is prevented, achieving simultaneous high $CO_2$ conversion and high CO selectivity. The catalytic performance of this catalyst exceeds almost all reported values in RWGS, demonstrating that the hydroxylation of oxide supports can play an important role in developing high-performance supported catalysts.

## Results

### Formation and structure of the Ni-cluster/$TiO_2$ catalyst

By using the commercial anatase $TiO_2$ as a precursor (denoted by $TiO_2$-Ref1, Fig. 1a), hydroxylated $H_2Ti_3O_7$ was obtained as support for the Ni/$TiO_2$ catalyst[37,38]. The detailed synthesis process and the corresponding catalyst structure evolution were shown as follows. Firstly, during a hydrothermal treatment under highly concentrated NaOH (10 mol/L) with subsequent HCl washing, a tube-shaped $H_2Ti_3O_7$ phase (denoted by $TiO_2$-OH, Fig. 1b and Supplementary Fig. 1) was prepared as the result of $TiO_2$ reconstruction. Owing to the phase transformation, plenty of intrinsic −OH groups were identified on the surface of $TiO_2$-OH,

whereas very few −OH groups were present on the reference $TiO_2$-Ref1 (Supplementary Fig. 2). Secondly, 10 wt% Ni was loaded on $TiO_2$-OH by the deposition−precipitation method, and the obtained sample without air calcination was denoted by 10Ni/$TiO_2$-OH-UC (Fig. 1c). The catalyst was subsequently calcined in air at 600 °C. The hydroxylated $H_2Ti_3O_7$ underwent dehydration, leading to a phase change back to anatase $TiO_2$ (Supplementary Fig. 3). The attenuated total internal reflectance Fourier transform infrared spectroscopy (ATR-FTIR) test proved the maintenance of −OH groups on the surface even after calcination (Supplementary Fig. 4). Meanwhile, Ni was still well dispersed (denoted by 10Ni/$TiO_2$-OH, Fig. 1d). The residual surface −OH on the as-prepared 10Ni/$TiO_2$-OH catalyst improved the stability of isolated Ni atoms. The absence of Ni diffraction peaks in the X-ray diffraction (XRD) pattern for 10Ni/$TiO_2$-OH implied the very small crystalline size of Ni species (Supplementary Fig. 3). The high dispersion of Ni over 10Ni/$TiO_2$-OH was further confirmed by the atomic-resolution high-angle annular dark-field scanning transmission electron microscopy (HAADF-STEM) images and X-ray energy dispersive spectroscopy (EDS) elemental mapping results (Fig. 1d and Supplementary Fig. 5). The Fourier-transform extended X-ray absorption fine structure (EXAFS) results of 10Ni/$TiO_2$-OH showed no Ni−Ni bond, demonstrating that Ni species were dispersed on the $TiO_2$-OH support dominantly as single Ni atoms, though the presence of a small number of sub-nanometer Ni clusters could not be ruled out (Supplementary Fig. 6 and Supplementary Table 1). It was noteworthy that Ni single atoms on the catalyst surface were very sensitive to the radiation of high energy electron beam in TEM operating at 300 kV, and they would aggregate (<1 nm) with the increase of radiation time (Supplementary Fig. 7). Therefore, all the STEM images were collected within 10 s under small beam current. The schematic illustration of the generation process of 10Ni/$TiO_2$-OH is depicted in Fig. 1e.

Structure of the catalyst was further studied after the $H_2$ pretreatment and $CO_2$ hydrogenation (Fig. 2a). After the RWGS reaction (23%$CO_2$/69%$H_2$/8%$N_2$) at 600 °C, isolated Ni atoms were converted into Ni clusters with the average size of ~1 nm (Fig. 2b and Supplementary Fig. 8) and a few large-sized Ni particles (Fig. 2c and Supplementary Fig. 9) on the sintered $TiO_2$ support. The EDS-mapping results further indicated the good dispersion of Ni clusters, with Ni element appearing uniformly with Ti and O (Fig. 2d and Supplementary Fig. 10). Due to the strong metal-support interaction (SMSI)[39], Ni particles were

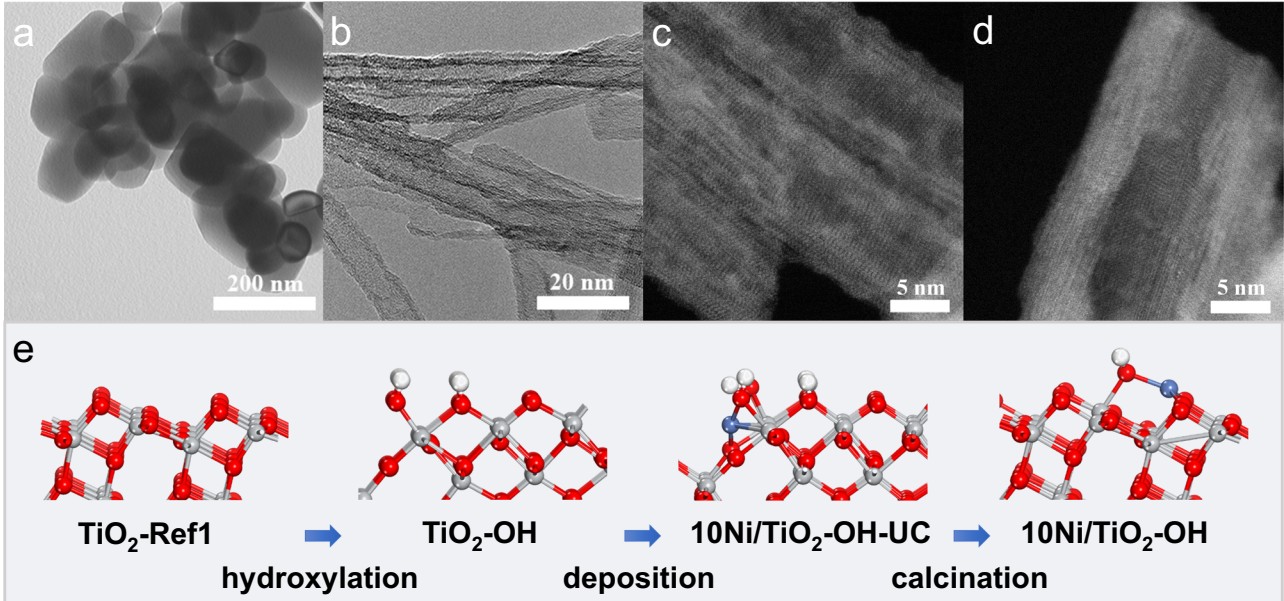

**Fig. 1 | Structural information of catalyst.** TEM images of **a** $TiO_2$-Ref1 and **b** $TiO_2$-OH samples. HAADF-STEM images of the **c** 10Ni/$TiO_2$-OH-UC catalyst and **d** 10Ni/$TiO_2$-OH catalyst. **e** Schematic illustration of the generation process for the 10Ni/$TiO_2$-OH catalyst.

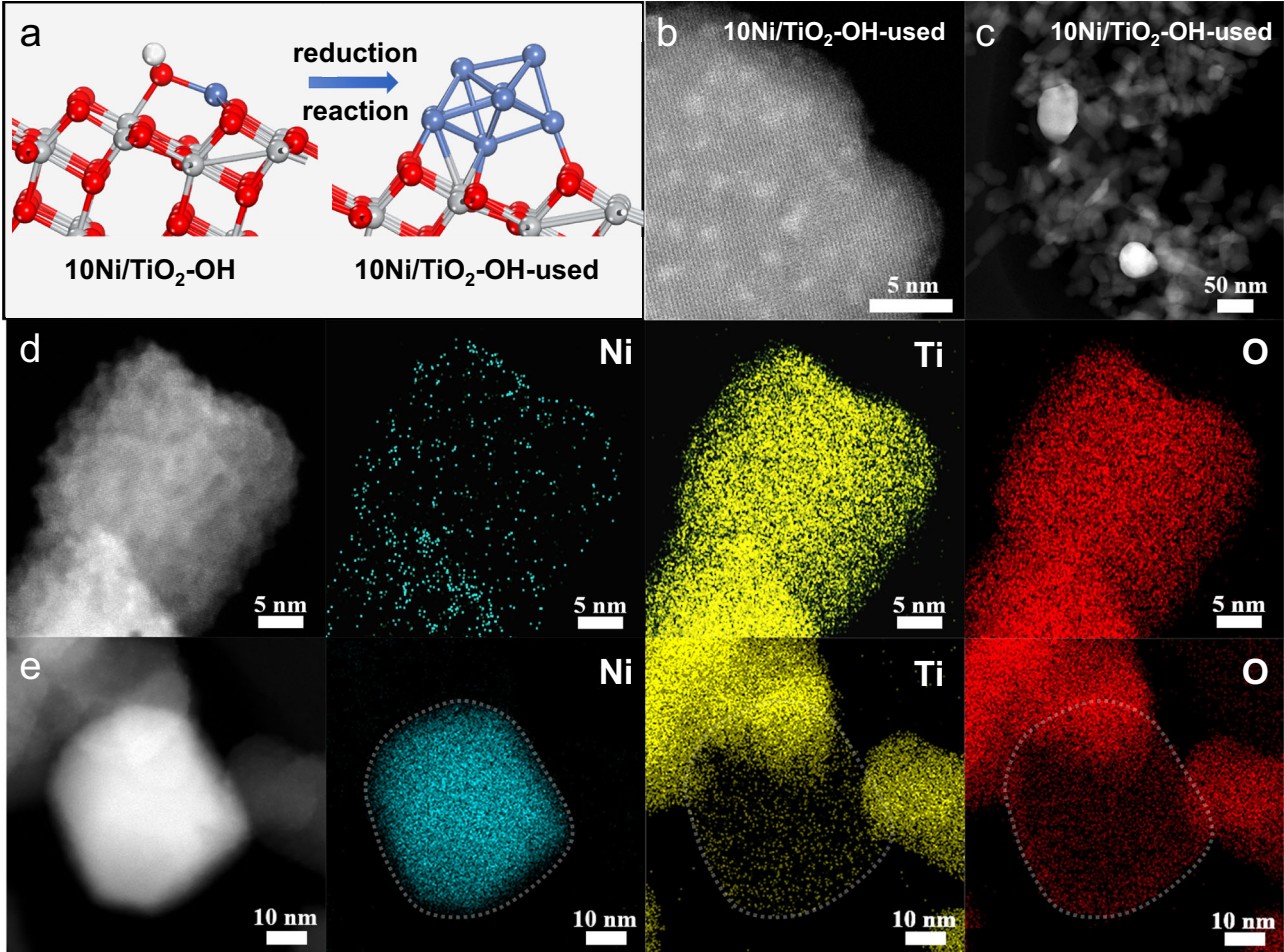

**Fig. 2 | Structural information of catalyst. a** Schematic illustration of the generation process for the used 10Ni/TiO$_2$-OH catalyst from the as-prepared 10Ni/TiO$_2$-OH catalyst. **b**, **c** HAADF-STEM images of the used 10Ni/TiO$_2$-OH catalyst. The EDS-elemental mapping results for the **d** Ni cluster and **e** Ni particle of the used 10Ni/TiO$_2$-OH catalysts.

covered by a thin TiO$_x$ overlayer (Fig. 2e and Supplementary Fig. 11), which has been widely reported in previous literatures[14,30,40]. The X-ray absorption near edge spectroscopy (XANES) of used 10Ni/TiO$_2$-OH indicated the reduction of Ni species during the H$_2$ pretreatment and the subsequent reaction process (Supplementary Fig. 6a). As illustrated by the EXAFS profiles (Supplementary Fig. 6b) and the 2-D contour plots wavelet transform (WT) results (Supplementary Fig. 6c, f), the obvious Ni–Ni contribution over used Ni/TiO$_2$-OH suggested the aggregation of single Ni atoms, which was consistent with the XRD result (Supplementary Fig. 12).

In contrast, fresh and used Ni/TiO$_2$-Ref1 catalysts exhibited distinctly different structures. Although the TiO$_2$-Ref1 support morphology maintained well during the synthesis process, Ni particles with an average size of ~9.6 nm were formed on the catalyst surface after the air calcination at 600 °C (Supplementary Fig. 13). After the pretreatment and reaction, this catalyst further underwent severe sintering, and the average size of Ni particles increased to ~27.6 nm (Supplementary Fig. 14). Large-size Ni particles was further confirmed by the element mapping (Supplementary Fig. 15). Interestingly, based on ex-situ EELS-mapping results for 10Ni/TiO$_2$-Ref1, no obvious TiO$_x$ overlayer was found on Ni particles (Supplementary Fig. 16), which might be caused by the limited reducibility of TiO$_2$-Ref1 (Supplementary Fig. 17). Based on above comprehensive experimental results, we provide a strategy for synthesizing Ni cluster catalysts by using hydroxylated TiO$_2$ as support. Compared to commercial TiO$_2$-Ref1 with poor ability to disperse Ni species,

hydroxylated TiO$_2$ with abundant –OH groups could effectively anchor Ni single atoms, thereby forming rich and durable Ni clusters even under harsh reaction conditions (600 °C and reducing atmosphere).

In order to explore the mechanism of the effect of –OH groups on anchoring single Ni atoms under the air calcination, we carried out density functional theory (DFT) calculations. We adopted the anatase TiO$_2$(101) surface to model the substrate. We noted that (101) was the most abundant surface of anatase TiO$_2$[41] that contained twofold coordinated O$_{2c}$ atoms and fivefold coordinated Ti$_{5c}$ atoms along the [010] direction (Supplementary Fig. 18). Supplementary Fig. 19 illustrated three types of hydroxylated TiO$_2$(101) surfaces, in which the adsorption configurations of the Ni single atom were explored. The stable structures are shown in Fig. 3a, with the calculated adsorption energies listed in Supplementary Table 2. When single Ni atom adhered to an –OH group located at the fivefold coordinated Ti$_{5c}$ atom (OH$_{5c}$/TiO$_2$, Supplementary Fig. 19a), the corresponding adsorption energies of Ni/OH$_{5c}$-TiO$_2$-I and Ni/OH$_{5c}$-TiO$_2$-II were higher than that on a bare anatase TiO$_2$(101) surface (−5.83 eV, −5.81 eV vs. −2.89 eV). Notably, these values were also higher than the calculated cohesive energy of bulk Ni (−5.83 eV, −5.81 eV vs. −5.47 eV), meaning that the interaction between the Ni atom and the OH$_{5c}$/TiO$_2$ substrate was stronger than the interaction between Ni atoms in their bulk phase. Bader charge analysis (Supplementary Table 2) showed that the single Ni atom in Ni/OH$_{5c}$-TiO$_2$-I and Ni/OH$_{5c}$-TiO$_2$-II exhibited considerable positive charges (+0.88 |

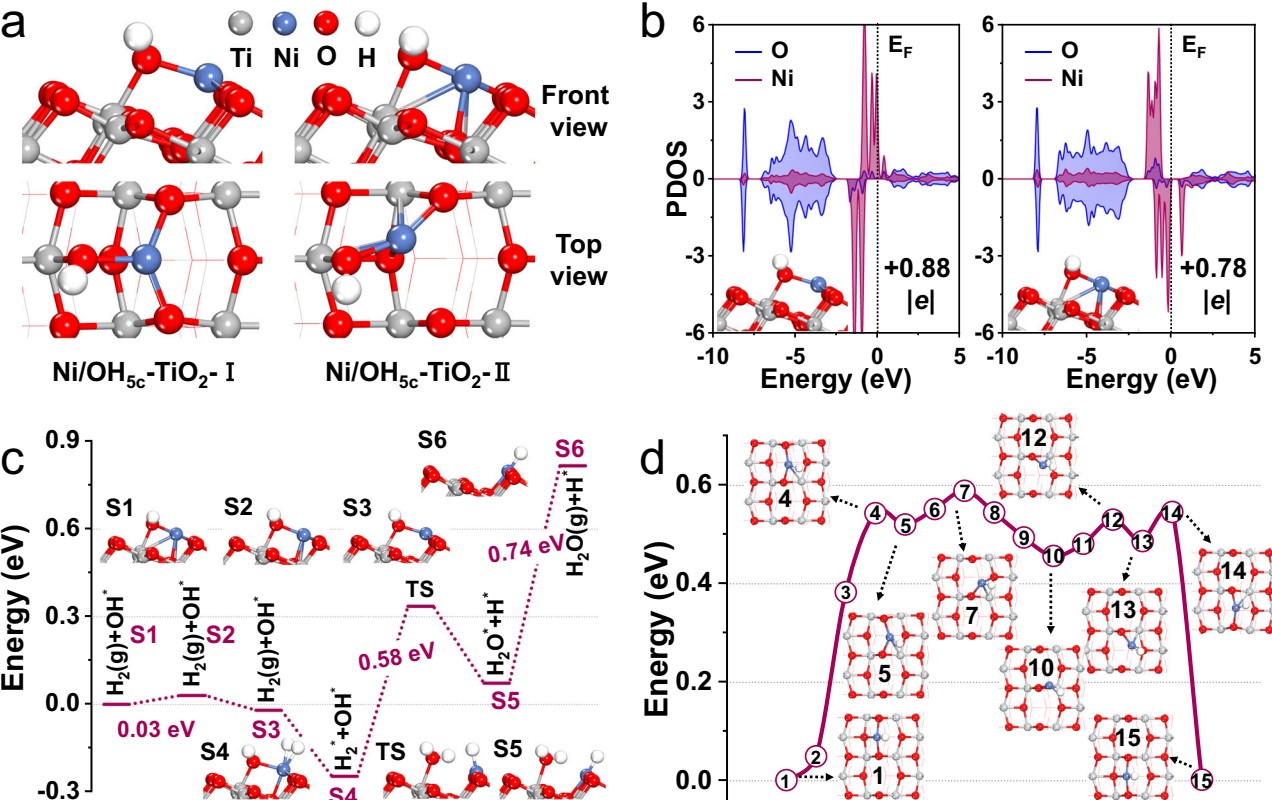

**Fig. 3 | Structural evolution of catalyst. a** Two most stable adsorption configurations of a single Ni atom on an $OH_{5c}/TiO_2$ surface (denoted as Ni/$OH_{5c}$-TiO$_2$-I and Ni/$OH_{5c}$-TiO$_2$-II, respectively), with the corresponding adsorption energies shown in Supplementary Table 2. **b** Projected electronic density of states (PDOS) of the single Ni atom and of the O atoms bonded with Ni in Ni/$OH_{5c}$-TiO$_2$-I (left) and Ni/

$OH_{5c}$-TiO$_2$-II (right). The Fermi level was set to 0 eV. **c** Potential energy diagram for the removal of −OH group and the generation of the Ni-H species. Here, TS represents the transition state, and S1 and S3 correspond to Ni/$OH_{5c}$-TiO$_2$-II and Ni/$OH_{5c}$-TiO$_2$-I, respectively. **d** Potential energy diagram for the diffusion of the Ni-H species on the TiO$_2$(101) surface.

$e|$ and +0.78 $|e|$), which enabled it to be stabilized via the electrostatic attraction with surrounding O atoms[42]. In addition, the projected electronic density of states of the Ni atom overlapped with that of the bonded O atoms (Fig. 3b), indicating that covalent interaction also existed between Ni and O. Hence, both electrostatic and covalent interactions within the Ni−O bonds contributed to the excellent stability of the adsorbed Ni atoms on the Ni/$OH_{5c}$-TiO$_2$-I and Ni/$OH_{5c}$-TiO$_2$-II models.

During the subsequent H$_2$ pretreatment and the RWGS reaction, the potential mechanism of the transformation process from isolated Ni atoms to clusters was also explored by DFT simulations. Here, both Ni/$OH_{5c}$-TiO$_2$-I and Ni/$OH_{5c}$-TiO$_2$-II were considered as the initial states to investigate the subsequent structural changes of the Ni single atom. Under an H$_2$ atmosphere, the −OH group in Ni/$OH_{5c}$-TiO$_2$-I could be easily removed by forming a water molecule with the H atom (Fig. 3c: S4 → TS → S5). Regarding Ni/$OH_{5c}$-TiO$_2$-II, although a direct removal of the −OH group (Supplementary Fig. 20: S2 → TS → S3) was a relatively hard process with an energy barrier of 1.06 eV, Ni/$OH_{5c}$-TiO$_2$-II could facilely transform to Ni/$OH_{5c}$-TiO$_2$-I with an energy barrier of only 0.03 eV (Fig. 3c: S1 → S2 →S3). It meant that the −OH removal in Ni/$OH_{5c}$-TiO$_2$-II could also occur easily, following the same pathway as that of Ni/$OH_{5c}$-TiO$_2$-I. Upon desorption of the water molecule (Fig. 3c: S5 → S6), a Ni−H species was left on the TiO$_2$(101) surface. This Ni−H species could easily diffuse on the TiO$_2$(101) surface, exhibiting an energy barrier of only 0.59 eV (Fig. 3d). As a comparison, the diffusion of an isolated Ni atom was also considered (Supplementary Fig. 21), and the corresponding energy barrier was calculated to be 1.03 eV. It could be seen that under reaction conditions, the H atom adsorbed on Ni could

promote its diffusion, which was reminiscent of the effect of CO on the diffusion of Pt adatoms on Fe$_3$O$_4$(001)[43]. The removal of the −OH group and the promotion of the Ni diffusion under the H$_2$ atmosphere well explained the agglomeration of Ni atoms to clusters observed after CO$_2$ hydrogenation.

## Catalytic performance of the Ni/TiO$_2$-OH catalysts with rich and stable Ni clusters

It has been reported that compared to large-size Ni particles, Ni clusters with poor electronic transfer to CO molecules can promote the desorption of carbonyl (CO*) species to form gaseous CO molecules[9]. Therefore, it can be expected that the obtained 10Ni/TiO$_2$-OH catalyst with abundant Ni clusters could exhibit high catalytic performance to catalyze the RWGS reaction. As shown in Fig. 4a, 10Ni/TiO$_2$-OH had high CO$_2$ conversion, which was comparable to the thermodynamic equilibrium limitation at 500 °C and 600 °C. However, due to the aggregated Ni particles, the CO$_2$ conversion of 10Ni/TiO$_2$-Ref1 was less than 10% even at 600 °C. Meanwhile, 10Ni/TiO$_2$-OH showed excellent CO selectivity, maintaining above 90% at all reaction temperatures, in clear contrast to the generally observed near 100% selectivity towards methane which is generally observed on nickel-based catalysts. In order to further compare the product selectivity of 10Ni/TiO$_2$-OH and 10Ni/TiO$_2$-Ref1 with distinctly different Ni structures, additional activity tests were performed. Through controlling GHSV, we tried to compare the product selectivity of 10Ni/TiO$_2$-OH and 10Ni/TiO$_2$-Ref1 under a similar CO$_2$ conversion. As shown in Fig. 4b, even when the GHSV was increased to 800,000 mL g$_{cat}^{-1}$ h$^{-1}$, the CO$_2$ conversion of 10Ni/TiO$_2$-OH was still much higher than that of 10Ni/TiO$_2$-Ref1 with a much lower GHSV of 20,000 mL g$_{cat}^{-1}$ h$^{-1}$. Under such test conditions,

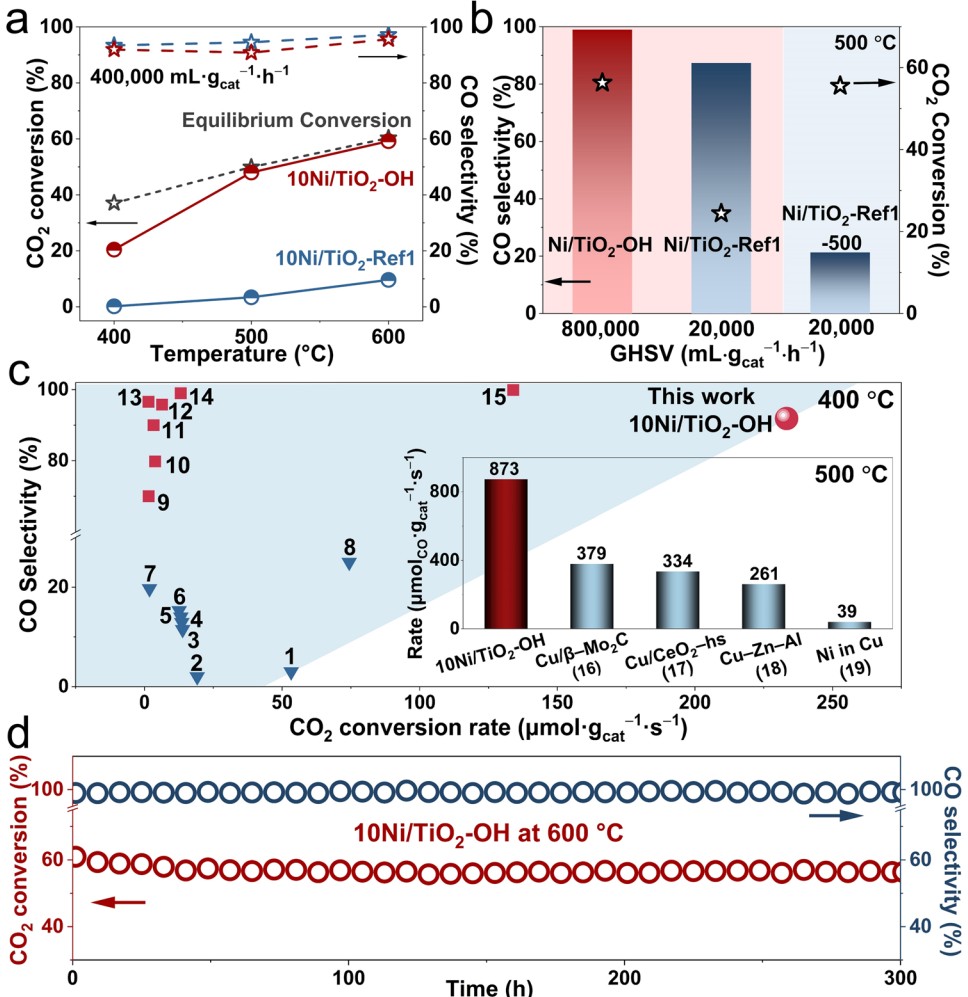

**Fig. 4 | Catalytic performance for CO₂ hydrogenation. a** The CO₂ conversion and CO selectivity for 10Ni/TiO₂-OH and 10Ni/TiO₂-Ref1 catalysts under the GHSV of 400,000 mL g$_{cat}^{-1}$ h$^{-1}$. **b** Comparison of CO selectivity at 500 °C within relative comparable CO₂ conversion rates for 10Ni/TiO₂-OH, 10Ni/TiO₂-Ref1 and 10Ni/TiO₂-Ref1 after calcination at 500 °C and H₂ pretreatment at 500 °C (10Ni/TiO₂-Ref1-500). **c** CO₂ conversion rates versus CO selectivity reported non-Cu based catalyst for CH₄ production (blue triangles, Sel$_{CO}$ < 50%) and CO production (red squares, Sel$_{CO}$ > 50%), and this work (red circle) at 400 °C. Inset: comparison of CO yield rates over 10Ni/TiO₂-OH catalysts and Cu-based catalysts at 500 °C. **d** The long-term stability of the 10Ni/TiO₂-OH catalyst tested for 300 h at 600 °C (GHSV = 800,000 mL g$_{cat}^{-1}$ h$^{-1}$).

the CO selectivity of 10Ni/TiO₂-Ref1 was slightly lower but still over 85% compared to that of 10Ni/TiO₂-OH, over 98%, which seemed to conflict with the previously reported results that large-size metal particles led to the high CH₄ selectivity[9]. To resolve this confusion, by reducing the air calcination temperature and the subsequent pretreatment temperature of 10Ni/TiO₂-Ref1 from 600 °C to 500 °C, we further synthesized another reference catalyst with relative smaller Ni particles than 10Ni/TiO₂-Ref1 and referred to as 10Ni/TiO₂-Ref1-500, in which Ni particle size was ~12.4 nm (Supplementary Fig. 22). As the particle size decreased, the catalytic activity was improved compared to 10Ni/TiO₂-Ref1, but it was still lower than that of 10Ni/TiO₂-OH with abundant Ni clusters (Supplementary Fig. 23). Importantly, at similar CO₂ conversion rates, the CO selectivity of 10Ni/TiO₂-OH (98.9%) was much higher than that of 10Ni/TiO₂-Ref1-500 (21.3%), indicating the selectivity difference between clusters and particles in catalyzing CO₂ hydrogenation (Fig. 4b). Based on above results, the relationship between the Ni size on TiO₂ and the product selectivity in atmospheric CO₂ hydrogenation could be clearly revealed. On the one hand, compared to highly dispersed Ni clusters, Ni particles induced the formation of CH₄. In addition, when the average size of Ni particle was larger than 27 nm, it was meaningless to discuss the product selectivity due to the very poor catalytic activity.

As reported, extensive efforts have been made to optimize the CO selectivity of non-Cu catalysts by regulating the catalyst structure[2,10–16]. However, due to the presence of an activity-selectivity trade-off, it was very challenging to achieve the combination of high activity and high selectivity (non-Cu-based catalysts ordered 1–15[13,23,44–48] in Fig. 4c and Supplementary Table 3). Obviously, 10Ni/TiO₂-OH successfully broke the trade-off between activity and selectivity with a high CO selectivity of 91.9% while maintaining an excellent reaction rate of 214.4 μmol g$_{cat}^{-1}$ s$^{-1}$ at 400 °C. Furthermore, compared to Cu-based catalysts, which were regarded as prevailing catalysts for catalyzing the RWGS reaction, 10Ni/TiO₂-OH with an excellent reaction rate of 873.0 μmol$_{CO}$ g$_{cat}^{-1}$ s$^{-1}$ and a high CO selectivity of 98.0% at 500 °C was also highly advantageous (Cu-based catalysts ordered 16–19[6,49,50] in inset of Fig. 4c and Supplementary Table 3). The apparent activation energy ($E_a$) and the corresponding CO₂ conversion and CO selectivity for 10Ni/TiO₂-OH were shown in Supplementary Fig. 24. Besides, the catalytic performance of 10Ni/TiO₂-OH was evaluated in different reaction atmospheres with various H₂ to CO₂ ratios. With the increase of the ratio of H₂ to CO₂, CO₂ conversion increased slightly (Supplementary Fig. 25a), suggesting this catalyst possessed a high catalytic activity over a wide input reaction gas composition. Moreover, it was found that the CO selectivity maintained over 90% even when the H₂ to

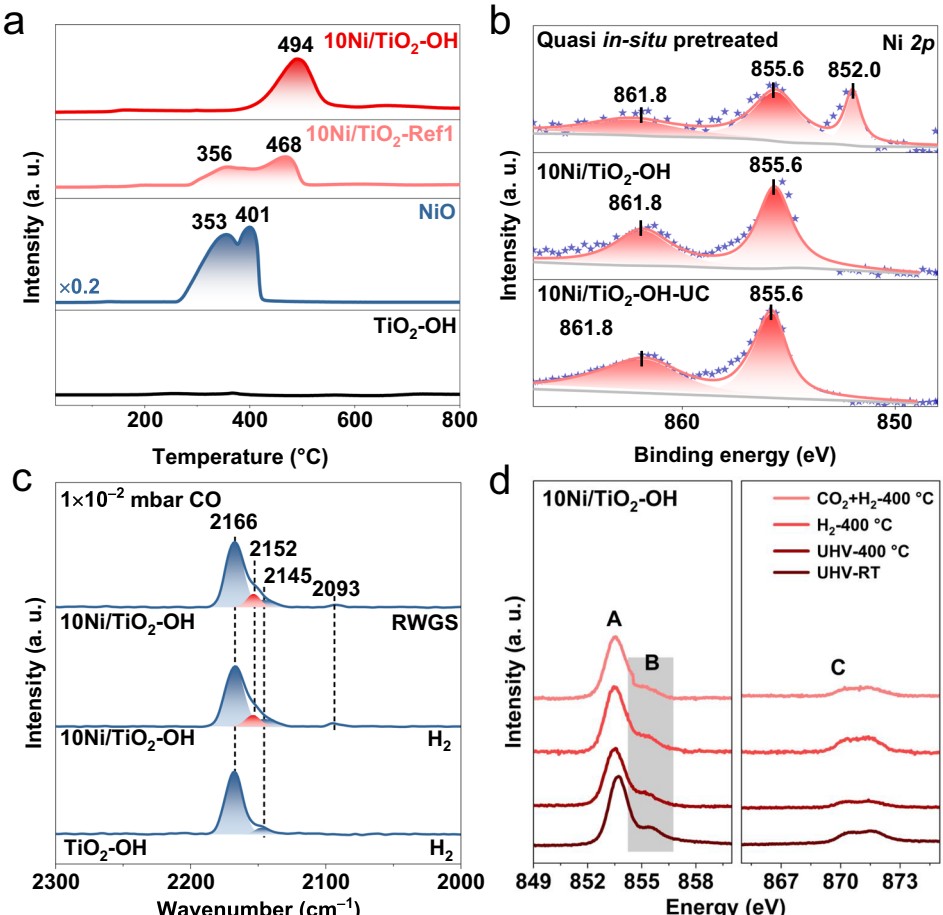

**Fig. 5 | Interfacial structure of Ni/TiO$_2$ catalyst. a** H$_2$-TPR profiles of the NiO, 10Ni/TiO$_2$-Ref1 and 10Ni/TiO$_2$-OH samples. **b** XPS spectra and corresponding fitting curves of Ni 2p in the 10Ni/TiO$_2$-OH-UC and 10Ni/TiO$_2$-OH catalysts. Quasi in situ XPS spectra and corresponding fitting curves of Ni 2p in 10Ni/TiO$_2$-OH catalyst after RWGS reaction at 600 °C for 3 h. **c** In situ infrared spectra recorded after exposing the TiO$_2$-OH after H$_2$ pretreatment and 10Ni/TiO$_2$-OH catalysts after H$_2$ pretreatment and RWGS reaction to CO at 130 K with $1 \times 10^{-2}$ mbar. **d** The in-situ Ni L-edge NAP-NEXAFS profile of the 10Ni/TiO$_2$-OH catalyst.

CO$_2$ ratio reached 4:1 (Supplementary Fig. 25b), which indicated that the methanation was indeed inhibited by this catalyst. Due to the harsh reaction conditions of the high-temperature RWGS reaction, reported supported catalysts are always severely deactivated by the catalyst sintering[51,52]. In this work, the long-term stability of 10Ni/TiO$_2$-OH was evaluated. As shown in Fig. 4d, under very harsh reaction conditions (600 °C and GHSV = 800,000 mL g$_{cat}^{-1}$ h$^{-1}$), 10Ni/TiO$_2$-OH remained stable with the CO$_2$ conversion more than ~55% and showed almost complete CO selectivity. Besides, 10Ni/TiO$_2$-OH also suggested a high start-up cool-down stability (Supplementary Fig. 26a). In contrast, under relatively milder reaction conditions, 10Ni/TiO$_2$-Ref1 lost almost all of its activity in only 2.5 h (Supplementary Fig. 26b), which was accompanied by the significant increase in particle size (Supplementary Fig. 27), further demonstrated the importance of suppressing the Ni sintering for reaction activity. Based on the above results, 10Ni/TiO$_2$-OH with rich Ni clusters efficiently broke the activity-selectivity trade-off of the RWGS reaction and achieved a combination of high activity, high CO selectivity and excellent stability. To the best of our knowledge, the catalytic performance of 10Ni/TiO$_2$-OH was unmatched by almost all other reported catalysts employed for the atmospheric high-temperature RWGS reaction.

The structure of the catalyst after 300 h of steady-state reaction at 600 °C was explored by HAADF-STEM. As shown in Supplementary Fig. 28, Ni clusters were still anchored on the catalyst surface. This result again emphasized the efficiency of −OH groups on the TiO$_2$-OH support for constructing active Ni structure.

## Strong interaction between the Ni cluster and TiO$_2$-OH

It has been established that catalyst structures and catalytic performances are largely determined by the MSI. Therefore, it is essential to reveal the interaction between Ni species and the TiO$_2$-OH support for 10Ni/TiO$_2$-OH, thereby understanding the stable interfacial structure and the molecule activation behavior. Firstly, we used the temperature-programmed hydrogen reduction (H$_2$-TPR) technique to investigate the redox properties of the 10Ni/TiO$_2$-OH catalyst. As shown in the H$_2$-TPR profile (Fig. 5a), pure NiO exhibited two reduction peaks centered at 352 °C and 401 °C, which could be attributed to the cascading reduction of Ni$^{2+}$ to Ni$^+$ and then to Ni$^0$. 10Ni/TiO$_2$-Ref1 showed similar reduction features to that of pure NiO, suggesting a very weak interaction between Ni species and TiO$_2$-Ref1. Compared to pure NiO, the increased reduction temperatures for 10Ni/TiO$_2$-Ref1 might be caused by the size difference of NiO. The reduction peak of the 10Ni/TiO$_2$-OH sample was centered at 494 °C, much higher than that of the above two samples, which indicated a higher energy barrier to break the Ni-O bond. By quantifying the consumed H$_2$ based on H$_2$-TPR results (Supplementary Fig. 29), it could be found that the reduction peaks of 10Ni/TiO$_2$-Ref1 were mainly from the NiO reduction to Ni since the actual hydrogen consumption was close to the theoretical value. This again confirmed the weak MSI between Ni species and TiO$_2$-Ref1. However, the actual H$_2$ consumption of 10Ni/TiO$_2$-OH was much higher than the theoretical H$_2$ consumption, suggesting that partial surface oxygen atoms of TiO$_2$-OH support were reduced to generate TiO$_x$ structure. According to previous literature, the formation of TiO$_x$ could cover the large-size metal

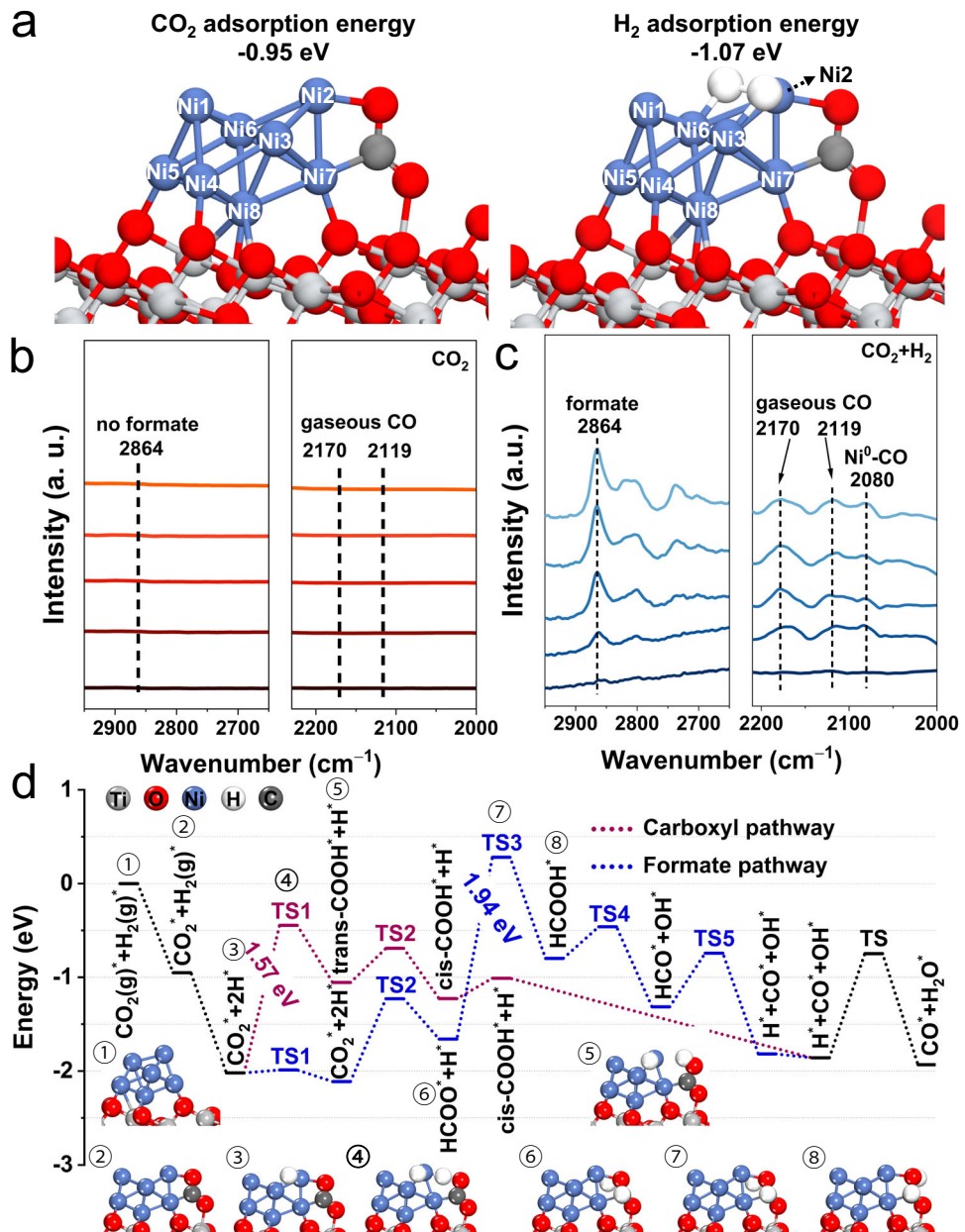

**Fig. 6 | Theoretical and experimental results for the reaction mechanism. a** $CO_2$ adsorption on $Ni_8$/$TiO_2$ system and $H_2$ adsorption on $CO_2$-$Ni_8$/$TiO_2$ system. **b**, **c** In situ diffused reflectance infrared Fourier transform spectroscopy (DRIFTS) spectra of 10Ni/$TiO_2$-OH catalyst during $CO_2$ treatment and reaction conditions ($CO_2$ + $H_2$) at 300 °C, respectively. **d** Potential energy diagram for the RWGS reaction on $Ni_8$/ $TiO_2$(101) via carboxyl and formate pathways. The illustration showed the structures of some intermediate and transition states. The complete reaction pathways are shown in Supplementary Figs. 38 (carboxyl) and 39 (formate). "TS" represents the transition state.

particles[53,54], so large-size Ni particles on 10Ni/$TiO_2$-OH were covered by $TiO_x$, while the same phenomenon was not found on 10Ni/$TiO_2$-Ref1. The above analysis of $H_2$-TPR results clearly revealed that $TiO_2$-OH with abundant −OH groups could induce the formation of the enhanced interaction between Ni and $TiO_2$-OH, promoting the sintering-resisting capacity of Ni clusters on the $TiO_2$ surface. Actually, in addition to anchoring Ni species, such a strong interaction stabilized the $TiO_2$. Unlike pure $TiO_2$-OH supports seriously sintered after the air calcination at 500 °C (Supplementary Fig. 30a, b), $TiO_2$ in 10Ni/$TiO_2$-OH maintained the tube-shape morphology well (Supplementary Fig. 30c, d).

The strong interaction between Ni and $TiO_2$ in 10Ni/$TiO_2$-OH was further verified by the following quasi in situ XPS, in situ infrared spectra (IR), and in situ near ambient pressure (NAP) near edge X-ray absorption fine structure (NEXAFS). From the Ni $2p$ XPS spectra

(Fig. 5b), the Ni $2p_{1/2}$ binding energy (B.E.) of as-prepared 10Ni/$TiO_2$-OH was at ~855.6 eV, which was attributed to the presence of $Ni^{2+2}$. After the RWGS reaction, besides the $Ni^{2+}$ species at ~855.6 eV, the $Ni^0$ species at ~852.0 eV also appeared[22], which suggested the reduction of $Ni^{2+}$ under the reducing atmosphere. The $Ni^{2+}$/($Ni^0$ + $Ni^{2+}$) ratio was calculated to be ~66%, indicating that even after the treatment by the reaction gas with highly concentrated $H_2$ of 69%, a majority of $Ni^{2+}$ species still existed on the catalyst surface. The presence of abundant $Ni^{2+}$ species represented the electronic metal-support interaction (EMSI) between Ni species and $TiO_2$, which prevented Ni species from agglomerating on the $TiO_2$ support surface caused by the easy reduction. Because the Ni species in 10Ni/$TiO_2$-OH with excellent CO selectivity featured a weak adsorption capacity for CO molecules, it was difficult to use CO molecules as probes to characterize the state of

Ni species by in situ DRIFTS at room temperature. In general, the decrease in temperature could improve the adsorption strength of metal species for CO molecules, so as to more clearly characterize the state of metal species[55]. Therefore, in situ IR spectra of CO adsorption were collected at a low testing temperature of 130 K with a pressure of $1 \times 10^{-2}$ mbar (Fig. 5c). Pristine $TiO_2$-OH pretreated with $H_2$ exhibited a main peak at 2168 $cm^{-1}$ with a shoulder peak at 2147 $cm^{-1}$, ascribed to the adsorption of CO on distinct exposed facets of $TiO_2$[56]. By contrast, there were two extra peaks in the Ni/$TiO_2$-OH catalyst. One at 2152 $cm^{-1}$ was assigned to CO linearly adsorbed on $Ni^{2+}$ sites[57,58]. In addition, the signal at 2090 $cm^{-1}$ likely corresponded to sub-carbonyl $Ni(CO)_x$ ($x = 2$ or 3) species, which featured a highly disordered structure and were attributed to amorphous Ni sites[59,60]. Similar results were observed as the CO partial pressure was reduced to $1 \times 10^{-7}$ mbar (Supplementary Fig. 31). The dominant $Ni^{2+}$-CO signal confirmed the presence of rich $Ni^{2+}$, which again presented the strong EMSI between Ni and $TiO_2$. Moreover, the sub-carbonyl species were weakly adsorbed, which was favored by CO removal, thereby facilitating the high CO selectivity of the RWGS reaction. Ni L-edge NAP-NEXAFS was further measured to detect the surface Ni species under various atmospheres. Auger electron yield (AEY) for Ni $L_3$ (853 eV to 856 eV) and $L_2$ (870 eV to 872 eV) was observed (Fig. 5d), which were related to $2p_{3/2}$ to $3d$ and $2p_{1/2}$ to $3d$ transition, respectively. For 10Ni/$TiO_2$-OH, the relatively stable Ni L edge signals suggested the high stability of $Ni^{2+}$ even under $H_2$ and RWGS reaction flows at 400 °C, which further indicated the EMSI between Ni and the support. In comparison, 10Ni/$TiO_2$-Ref1 showed a significant change in intensity at 855.5 eV under the $H_2$ atmosphere at 300 °C, demonstrating that $Ni^{2+}$ was easily reduced to $Ni^0$ (Supplementary Fig. 32). The surface structure of the $TiO_2$ support over 10Ni/$TiO_2$-OH was also explored by XPS and NAP-NEXAFS. The Ti $2p$ XPS spectra exhibited that even after the treatment by the RWGS reaction flow at 600 °C, only $Ti^{4+}$ species but no obvious $Ti^{3+}$ signal could be found (Supplementary Fig. 33a). The in situ Ti L-edge NAP-NEXAFS profile of 10Ni/$TiO_2$-OH also did not detect the formation of $Ti^{3+}$ under $H_2$ and reaction flows (Supplementary Fig. 33b). Based on above results, we speculated that even though the MSI between Ni and $TiO_2$ facilitated the partial reduction of $TiO_2$, the limited concentration of surface $Ti^{3+}$ sites were below the detection line of XPS and in situ NEXAFS techniques. The created $Ti^{3+}$ might be moved into the bulk phase of anatase $TiO_2$[61].

## The proposed reaction mechanism of $CO_2$ hydrogenation catalyzed by Ni/$TiO_2$-OH

To identify the potential reaction mechanism of $CO_2$ hydrogenation to CO catalyzed by this highly performed 10Ni/$TiO_2$-OH catalyst, we first evaluated the $CO_2$ adsorption ability through the temperature-programmed desorption of $CO_2$ ($CO_2$-TPD) (Supplementary Fig. 34). Compared to 10Ni/$TiO_2$-Ref1 with weak capacity to adsorb $CO_2$, two strong $CO_2$ desorption peaks appeared over 10Ni/$TiO_2$-OH. By comparing the result of Ar-TPD, the first feature at low temperatures could be ascribed to chemisorbed $CO_2$ molecules, and the latter one was related to the decomposition of surface carbides. The strong $CO_2$ desorption signal indicated that compared to 10Ni/$TiO_2$-Ref1, rich Ni cluster-$TiO_2$ interfaces might serve as crucial sites for absorbing $CO_2$. Subsequently, DFT calculations were performed to further understand the adsorption and activation of $CO_2$ on this Ni cluster catalyst through a catalyst model of $Ni_8$/$TiO_2$ surface ($Ni_8$/$TiO_2$ model seen Supplementary Fig. 35). It should be noted that because no distinct $Ti^{3+}$ signal was detected on the catalyst surface, we did not consider the presence of oxygen vacancies in this constructed catalyst model. As shown in Fig. 6a, $CO_2$ was easily adsorbed at the Ni cluster-$TiO_2$ interface with an adsorption energy of −0.95 eV, and the $Ni_8$ cluster became ship-shaped. Meanwhile, $H_2$ was spontaneously dissociated into two H* on the Ni cluster. This result declared that the Ni cluster/$TiO_2$ interface, which was ultra-

stable in the RWGS reaction, could be considered as a significant active structure to activate reactant molecules.

Moreover, we also explored the $CO_2$ hydrogenation mechanism of 10Ni/$TiO_2$-OH. The mechanisms of the RWGS reaction have been categorized into two types: redox mechanism and associative mechanism. The dissociation experiment of $CO_2$ was conducted to investigate the reaction pathway. After the $H_2$/Ar pretreatment at 600 °C, the $CO_2$/Ar mix gas was injected into the reactor after the sample was cooled to room temperature. However, there was no generation of CO and consumption of $CO_2$ during the subsequent warming process, which indicated that CO might not be produced by the direct dissociation of $CO_2$ on the 10Ni/$TiO_2$-OH catalyst (Supplementary Fig. 36a). In contrast, the temperature-programmed surface reaction (TPSR) results showed that the $CO_2$ signal gradually decreased start from 300 °C, accompanied by the increase of CO signal, demonstrating that the dissociation of $CO_2$ into CO required the assistance of $H_2$ (Supplementary Fig. 36b). By combining the results of the $CO_2$ dissociation experiment and TPSR, it could be inferred that $CO_2$ activation likely occurred through an associative intermediate pathway rather than the redox mechanism. Similar results were revealed by in situ diffuse reflectance infrared Fourier transform spectroscopy (DRIFTS). After $CO_2$ injection, only carbonate species appeared, without signal of formate and CO signal, which further proved that it was difficult for $CO_2$ to be directly dissociated to CO (Fig. 6b and Supplementary Fig. 37a). Subsequently, $H_2$ flow was introduced to the 10Ni/$TiO_2$-OH catalyst, a new feature corresponding to formate species was identified at 1606 $cm^{-1}$ [61], while it was absent with $N_2$ purging on the surface of the catalyst (Supplementary Fig. 37b). This result suggested that the reaction intermediate was generated in the presence of $H_2$ during the $CO_2$ dissociation. Moreover, the simultaneous injection of $CO_2$ and $H_2$ also promoted the formation of formate intermediates along with the gaseous and adsorbed CO signal (Fig. 6c), which was consistent with the TPSR results. DFT calculations were performed to further explore the associative mechanisms by using the $Ni_8$/$TiO_2$ catalyst model (Supplementary Fig. 35). Here, the carboxyl and the formate pathways were both taken into account. As shown in Fig. 6d and Supplementary Figs. 38 and 39, we presented the energy profiles of these two reaction pathways. It could be seen that the rate-determining steps of these two pathways were the formation of COOH* (carboxyl pathway) and HCOOH* (formate pathway), and the corresponding energy barriers were 1.57 eV and 1.94 eV (carboxyl: TS1; formate: TS3), respectively, indicating that the carboxyl pathway was kinetically favorable. Notably, the COOH* intermediate could easily decompose into CO* and OH*, while on the contrary, HCOO* was difficult to further hydrogenated and converted into products. This suggested that COOH* was a more active intermediate than HCOO*, which was in consistent with the previously reported spectator role of formate[62].

## Discussion

For heterogeneous catalytic reactions, the conversion of reactants and the selectivity of products often change largely along with the evolution of catalyst structures. Therefore, the construction of dense and stable active structures has always been at the core of the catalyst design. In this work, the ability of the commercial $TiO_2$ to anchor active metals is greatly enhanced via the hydroxylation, which guarantees that highly loaded Ni species were anchored dominantly as single atoms. The strong interaction between intrinsic −OH groups and highly dispersed Ni species was proven to play an essential role in the formation of sintering-resisting clusters in the RWGS reaction. Due to the strong MSI between Ni clusters and $TiO_2$ induced by the surface −OH, the formed rich Ni/$TiO_2$ interfaces not only promoted the $H_2$-assisted $CO_2$ dissociation but also suppressed the formation of $CH_4$, thereby breaking the activity-selectivity trade-off of the RWGS

reaction. This work provides a strategy to design high-performance supported catalysts for heterogeneous reactions with harsh conditions by constructing hydroxylated oxides with enriched −OH groups.

## Method

### Preparation of $TiO_2$ tube precursor

All of the chemicals applied to our experiments were of analytical grade and were used without further purification or modification. The $TiO_2$ tube precursor was prepared by hydrothermal method in Teflon-lined stainless-steel autoclaves. For a typical synthesis of support, 28 g of NaOH (Sinopharm) was dissolved in 70 mL of deionized water (10 mol/L). After that, 2 g of anatase $TiO_2$ (Macklin, particle size = 100 nm, denoted as $TiO_2$-Ref1) was added into the above solution with stirring for 1 h at room temperature. Then the mixture was heated at 130 °C for 24 h. After the hydrothermal synthesis, the fresh precipitates were filtered and washed with deionized water and 2 L of 0.2 M HCl (Sinopharm) aqueous solution followed by deionized water. The resulting material was dried in air at 80 °C for 18 h and then ground in a mortar. Hereafter, the $TiO_2$ tube precursor was denoted as $TiO_2$-OH.

### Preparation of Ni catalysts

In a typical deposition-precipitation (DP) method, 0.5 g $TiO_2$-OH powders were added to 25 mL deionized water at room temperature under vigorous stirring. 4.28 mL of 0.1 mol/L $Ni(NO_3)_2 \cdot 6H_2O$ (Sinopharm) were added to the above suspension dropwise. The pH value of the solution was kept at ca. 9 with the assistance of $Na_2CO_3$ (Macklin) during the whole course. After stirring at room temperature for 30 min, the precipitates were further aged at room temperature for 1 h. Then, they were purified by suction filtration with deionized water (1 L) at room temperature. Finally, the product was dried at 70 °C for 10 h and then calcined in air at 600 °C for 4 h[24,63]. In our report, the nickel-titanium samples are denoted as $x$Ni/$TiO_2$-OH, where $x$ is the nickel content in weight percent. The referenced catalyst was prepared by the same method only except that the calcination temperature was 500 °C was denoted as 10Ni/$TiO_2$-Ref1-500. The referenced catalyst was also prepared by the DP method as above just with the other purchased commercial $TiO_2$ (denoted as $TiO_2$-Ref1).

### Scanning transmission electron microscopy (STEM) characterization

High-angle annular dark-field (HAADF) STEM images, X-ray EDS spectra and elemental mappings, and EELS measurements were obtained from a Thermo Scientific Themis Z microscope equipped with a probe-forming spherical-aberration corrector at an operating voltage of 300 kV (Analytical Instrumentation Center of Hunan University).

### X-ray photoelectron spectroscopy (XPS)

The XPS measurements were carried out at a Thermo scientific ESCA-LAB Xi+ XPS spectrometer from Thermo Fisher. Quasi in situ XPS experiment was carried out on the same instrument. The spectrums of Ni 2$p$, Ti 2$p$, C 1$s$, and O 1$s$ were obtained after 3 h of reaction in 15% $CO_2$/ 30% $H_2$/$N_2$ atmosphere at 600 °C. The C 1$s$ signal located at 284.8 eV was used to calibrate each spectrum for accurate binding energies[24].

### Catalytic tests

The catalytic performance evaluation was tested in a fixed-bed flow reactor under a gas atmosphere of 23% $CO_2$/69% $H_2$/$N_2$ (66.7 mL min$^{-1}$, Deyang Gas Company, Jinan) at 1 bar total pressure[24]. Before activity test, 10 mg catalysts (20–40 mesh) diluted with 90 mg inert $SiO_2$ were activated by 5% $H_2$/Ar at 600 °C for 0.5 h followed by switching to the feed gas for testing. The 10Ni/$TiO_2$-Ref1-500 catalyst was activated by 5% $H_2$/Ar at 500 °C for 0.5 h followed by switching to the feed gas for testing. The test temperature ranges from 400 °C to 600 °C. Before the analysis of gas products, the RWGS reaction needs to stabilize for

1 h at each test temperature. The gas products were analyzed by using an on-line gas chromatograph equipped with a thermal conductivity detector (TCD). $CO_2$ conversion and CO selectivity were calculated using the following equations:

$$X_{CO_2} = \frac{n_{CO_2}^{in} - n_{CO_2}^{out}}{n_{CO_2}^{in}} \times 100\% \qquad (1)$$

$$S_{CO} = \frac{n_{CO}^{out}}{n_{CO}^{out} + n_{CH_4}^{out}} \times 100\% \qquad (2)$$

where $n_{CO_2}^{in}$ is the concentration of $CO_2$ in the reaction stream, and $n_{CO_2}^{out}$, $n_{CO}^{out}$, $n_{CH_4}^{out}$ are the concentrations of CO, $CO_2$, $CH_4$ in the outlet. For all catalysts, the $E_a$ was measured by using the same reactor for catalytic performance above. Appropriate amounts of catalysts diluted with inlet $SiO_2$ were used in the kinetics experiments. In order to obtain accurate kinetics data, the catalysts need to be first treated with reactive gas for an hour at 600 °C. During the kinetic test, the $CO_2$ conversion remained between 5% and 15% by changing the gas flow rate. In order to minimize the effect of the external diffusion, for a reaction rate at 500 °C, 3 mg of catalyst and reaction gas flow of 120 mL/min were used, and $CO_2$ conversion was 13.0% with CO selectivity of 98.07%. In order to better track the deactivation behavior, we used a much higher GHSV (800,000 mL $g_{cat}^{-1}$ h$^{-1}$) in the long-term stability test. Due to the poor CO selectivity of the reference catalyst, the rate of $CO_2$ conversion was used for the reaction rate at 400 °C, while the rate of CO production was used for the almost complete CO selectivity of the reference catalyst at 500 °C.

### Theoretical calculations

Details of the computational methods and the calculation model are put in the Supplementary Materials.

## Data availability

The main data supporting the findings of this study are available within the article and its Supplementary Information. Additional data are available from the corresponding authors upon request. Source data are provided with this paper.

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

## Acknowledgements

This work was financially supported by the National Science Fund for Distinguished Young Scholars of China (22225110, C.-J.J.), the National Key Research and Development Program of China (2021YFA1501103, C.-J.J.), the National Science Foundation of China (22075166, C.-J.J. and 22271177, W.-W.W.), the Science Foundation of Shandong Province of China (ZR2023ZD21, C.-J.J.), the Young Scholars Program of Shandong University (W.-W.W.), EPSRC (EP/P02467X/1 and EP/S018204/2, F.-R.W.), Royal Society (RG160661, IES\R3\170097, IES\R1\191035, IEC \R3\193038, F.-R.W.). We acknowledge SPring-8 (Japan) for the XAFS experiments conducted under proposal No. 2021A1387 and Dr. Hiroyuki Asakura from Kyoto University for helping with the XAFS measurement. We thank the Center of Structural Characterizations and Property Measurements at Shandong University for the help on sample characterizations.

## Author contributions

C.-J. Jia and F. R. Wang supervised the work; C.-X. Wang, X.-M. Lai, X.-H. Liu, X.-P. Fu and C.-J. Jia designed the experiments, analyzed the results, and wrote the manuscript; J.-Y. Li and Q. Fu performed the DFT simulations; C.-X. Wang, X.-M. Lai and W.-W. Wang performed the in situ DRIFTS, in situ Raman, and quasi in situ XPS; C.-X. Wang, and X.-M. Lai performed the catalysts preparation, catalytic tests and the TPR tests; C. Ma performed the aberration-corrected HAADF-STEM measurements and analyzed the results; H. Gu, X.-H. Liu and F. R. Wang performed the XAFS experiments and analyzed the data.

## Competing interests

The authors declare no competing interests.
