## [Peer Review File · Nature Communications]

Hydroxylated TiO₂-induced high-density Ni clusters for breaking the activity-selectivity trade-off of CO₂ hydrogenationREVIEWER COMMENTS

Reviewer #1 (Remarks to the Author):

I have read the article with interest but have some reservations, which are outlined below.

The catalysis part is the most convincing/interesting: Figure 4 clearly shows that Ni/TiO₂-OH is performing differently with respect to all other cats, and in particular to the Ni/TiO₂ reference.

The authors then try to explain this in terms of suppressed SMSI, but I do not see convincing evidence for this. Specifically, the claim "The coating phenomenon of Ni by TiO₂ due to the SMSI effect is avoided" is not corroborated by evidence. Many other possible hypotheses can be made as to why this catalyst behaves differently.

For example, based on what people have learned, the opposite might even be true: that the SMSI is stabilized in this case and not removed under reaction conditions as on Ni/TiO₂. The authors report TEM images that indicate formation of small clusters of Ni (plus some very big Ni particles). Based on the current understanding of structure sensitivity, smaller clusters tend to be active for RWGS more than methanation, so size effects could be the reason for the different selectivity, with SMSI being similar in the two systems.

The DFT part seems a bit off-topic too, seemingly discussing the stability of the "single atom" Ni system.

Figure 5 is the least convincing, and I believe the scheme in Figure 5d-e is not backed by evidence.

Hence, I recommend major revisions, which should further substantiate the findings reported by experimental evidence, before I can make a proper recommendation to reject or accept this work.

Reviewer #2 (Remarks to the Author):

Authors report the frustration of SMSI in the Ni/TiO₂ system by pre-functionalizing the support with OH before Ni deposition. The designed catalyst shows higher conversion than the catalyst without OH for CO₂ hydrogenation. Further, CO selectivity of Ni/TiO₂-OH was higher than using other supports (Al₂O₃, CeO₂). The work is interesting, both major revisions are needed before it can be reconsidered for acceptance.

- Fig.2 is missing, making it difficult to assess the interpretation of data.
- The callout of Figures in SI is messy and makes the reader run through pages to follow the story.
- Particle size distributions are needed to assess the precision of the synthesis methods.
- The existence/non-existence of SMSI relies heavily on localized STEM images, which can be misleading. Authors need to corroborate the existence/non-existence of SMSI via CO or N₂O chemisorption and/or DRIFTS, as well as compare the density of sites for the hydroxylated and non-hydroxylated sample before and after reaction.

- Authors claim that 10Ni/TiO₂-Ref1 is underperforming compared to 10Ni/TiO₂-OH catalyst because of SMSI; however, 10Ni/TiO₂-Ref1 has much larger Ni particles and thus less sites exposed, therefore lower conversion is not surprising. Authors need to compare performances on the basis of exposed Ni sites.
- For 10Ni/TiO₂-Ref1, were Ti³⁺ species were observed in XPS? thus supporting SMSI.

Reviewer #3 (Remarks to the Author):

In this manuscript the authors report the catalytic performance of CO₂ reduction to CO by 1 nm Ni clusters supported on hydroxylated TiO₂. They ascribe the CO selectivity to the inhibited SMSI effect. The formation mechanism of SMSI effect is indeed a critical but also challenging issue in heterogeneous catalysis. This is a good fundamental point to discuss, but there are some issues that should clarify. The language of the manuscript is not professional, and should be polished thoroughly.

- 1) For CO₂ hydrogenation, Ru and Ni are catalysts for CH₄ production; however, SMSI, decreasing particle size of metals, and increasing temperature will all increase CO selectivity. In this work the reaction temperature is 500 °C, how to exclude the effects of TiO_x overlayers. How to consider the effects of particle size and temperature?
- 2) According to the catalytic condition, GHSV should be 400,000 mL·gcat⁻¹·h⁻¹ not 800,000 mL·gcat⁻¹·h⁻¹.
- 3) Figure caption of Fig. 4a is not correct.
- 4) The scheme of SMSI in Fig. 5e is not reasonable. As the mechanism of SMSI is quite complicated, the author should provide solid supports for their guess and conclusions.
- 5) line 50: should be driving force ; what is meaning of the very Ni large particle in line 114.

Reviewer #4 (Remarks to the Author):

The authors aimed to address the challenge of finding efficient and cost-effective catalysts for the reverse water gas shift (RWGS) reaction, crucial for CO₂ hydrogenation. Although Ni nanoparticles are excellent catalysts for hydrogenation, they are prone to encapsulation through classical strong metal-support interaction (SMSI) with sub-oxide species at high reduction temperatures, potentially limiting their use in the RWGS reaction. To overcome this, the authors claim to create SMSI-resistant Ni-clusters on hydroxylated TiO₂. The authors claim to deposit Ni in single-atom form on the surface and then transforming them into clusters after hydrogen treatment, thereby they avoid the coating or encapsulation of Ni by TiO_{2-x} also known as the SMSI effect. This approach enabled enhanced catalytic performance and stability at temperatures as high as 600°C for the RWGS reaction.

While I do believe the authors have interesting work here, I have a few conceptual issues with the work. Firstly, it is well-known that encapsulation by SMSI occurs less on nanoclusters than on nanoparticles due to the concentration difference of phases. I.e., typically, due to the concentration

difference which causes phase-diffusion of sub-oxide species, metal clusters are actually much less susceptible to encapsulation by “reducible” MOs. So the authors claim to have come up with a new way to avoid something, which in fact was already well-known. Secondly, the authors fail to properly characterize their TiO₂-OH support. TiO₂ itself is active towards RWGS, and it may very well be that the functionalization created a change in the electronic/geometric structure of the support itself which is responsible for the observed changes, which should have been assessed; the electronic structure of the support post-treatment, and as well proper catalytic blank experiments should have been performed. I.e., proof is lacking where the additional activity/stability actually comes from. It could be a particle size effect of the metal, an electronic/geometric effect of the support itself, the availability of different sites, etc. Secondly, I have issues with the nomenclature; “SMSI-resistant”. MSI is a very broad phenomenon and I guess the authors are only referring to encapsulation/sub-oxide species covering? They should specifically discuss this aspect then. Indeed of course clusters by definition have a much higher MSI effect, and actually probably that is what is causing the high activity, the fact that the nanoclusters indeed very much feel the metal support interaction therefor this terminology is at its best confusing and at its worst incorrect. Thirdly, the authors claim that SMSI should by default be avoided, and miss relevant literature showing that this is actually not always the case.

All in all, I believe that the hydroxylation approach gave the authors interesting results, but the novelty of the findings is arguable (both from the encapsulation side, and from the approach side). Furthermore, the observed effects are not properly characterized/corroborated. Thus, I am of the opinion that this manuscript is better suited for a different journal.

Specific comments below:

“The Ni atoms were deposited 78 on the hydroxylated TiO₂ in single-atom form and subsequently transformed to clusters after hydrogen 79 treatment” “Ni 94 atomically anchored on the TiO₂-OH supports via a further calcination at 600 °C (denoted by 95 10Ni/TiO₂-OH, Fig. 1d).”

Proof is lacking to claim “atomic” sites, or single atoms. These are extremely difficult to properly characterize, and a few HAADF-STEM images are not enough to claim this, and X-ray EDS should actually even not show atomically dispersed sites; but rather suggests clusters? Also, it is said that XRD and XAFS show co-presence; but how can a technique it prove co-presence if both are bulk/ensemble average techniques?

Also, characterization of the reaction mechanism/proving that extra activity comes from the stabilization (electronic effect of the support on the Ni clusters) and not from the defect sites in TiO₂ which itself is also active in RWGS?

“However, Ni 39 nanoparticles are very susceptible to strong metal-support interaction (SMSI) with reducible oxides, 40 causing the encapsulation of Ni particles at high temperatures with strong reductive atmosphere.11–13” -- Indeed, nanoparticles, but clusters much less. You state it yourself: “Larger metal nanoparticles with a higher surface energy are more inclined to be encapsulated, when the minimization of surface free energy of metal nanoparticles is the major driven force.16-19

“Therefore, the SMSI effect in supported Ni catalysts prevents the activation of reactant molecules at 42 the active metal sites and restricts its application in the RWGS reaction.11,14,15” -- actually no, not necessarily, as SMSI can be used to tune selectivity. See relevant literature: 10.1126/science.adf6984, <https://doi.org/10.1038/s41467-019-12858-3>

“uncovered by TiO₂ under RWGS reaction conditions,” -- based on what proof?

“In our discovery, this was also proven by the fact that the larger Ni particles were encapsulated, while the Ni clusters were not. “ -- but it was already known, that NPs are more prone; why do the authors claim this is something new, that it is a “discovery”?

The reaction rate was 10 times higher, which suggests differing mechanisms; further suggesting that the actual underlying reason for the observed differences must better be corroborated (e.g., size vs MSI effect vs defect sites on treated TiO₂).

There is no blank with TiO₂-OH; how can the authors say their work performs better when in theory could simply be the TiO₂ which becomes more active? The TiO₂ (tube) precursor formation could severely affect the electronic properties of the TiO₂, and its effect should be evaluated.

“we developed a strategy to fabricate SMSI-resistant Ni clusters by reducing intrinsic hydroxyl-anchored isolated Ni atoms on the TiO₂ surface.” -- is it that, or is it simply the particle size preventing encapsulation which was known already?

Grammatical issues:

work pave a way -- paves

efficient and selective catalyst -- catalysts

H-H cracking ability -- affinity for H₂ dissociation

multi-molecule participated reaction -- rephrase

developed a approach -- an approach

Ni single atom could transform into clusters -- single Ni atoms could transform into clusters

Responses to the Reviewers' Comments and the Corresponding Revisions

Reviewer #1:

Comment: *I have read the article with interest but have some reservations, which are outlined below.*

Response: Thanks for reviewer's valuable comments. According to reviewer's comments and suggestions, we have carefully revised and supplemented the manuscript point by point.

Comment 1: *The catalysis part is the most convincing/interesting: Figure 4 clearly shows that Ni/TiO₂-OH is performing differently with respect to all other cats, and in particular to the Ni/TiO₂ reference.*

The authors then try to explain this in terms of suppressed SMSI, but I do not see convincing evidence for this. Specifically, the claim "The coating phenomenon of Ni by TiO₂ due to the SMSI effect is avoided" is not corroborated by evidence. Many other possible hypotheses can be made as to why this catalyst behaves differently.

For example, based on what people have learned, the opposite might even be true: that the SMSI is stabilized in this case and not removed under reaction conditions as on Ni/TiO₂. The authors report TEM images that indicate formation of small clusters of Ni (plus some very big Ni particles). Based on the current understanding of structure sensitivity, smaller clusters tend to be active for RWGS more than methanation, so size effects could be the reason for the different selectivity, with SMSI being similar in the two systems.

Response: Thanks for reviewer's valuable comments. According to reviewer's valuable comments and suggestions, we further explore the relationship between strong metal-support interaction (SMSI), size effects and catalytic performances. The detailed discussions are shown below.

1. The structure of Ni particles on the surface of the Ni/TiO₂-Ref1 catalyst.

The corresponding HRTEM images, HAADF-STEM images and EDS-elemental mapping results have been supplemented to investigate the structure of Ni particles on the surface of the Ni/TiO₂-Ref1 catalyst. As shown in Figure R1, large-sized Ni particles could be observed on the TiO₂-Ref1 support surface. And the average size of Ni particles was about 9.6 nm (the inset in Figure R1c). After the RWGS reaction at 600 °C, the Ni particles further aggregated into larger size of ~27.6 nm (Figure R2). The EELS-mapping results of two distinct Ni particle regions strongly confirmed that **on the surface of 10Ni/TiO₂-Ref1, large-sized Ni particles was not coated by TiO₂-Ref1 support** (Figure R3). The distinct-different structure (encapsulated by TiO₂ and exposed) of Ni particles on 10Ni/TiO₂-OH and 10Ni/TiO₂-Ref1 indicated that metal-support interactions in the two catalysts were not similar and Ni particles were more easily encapsulated by the TiO₂-OH support. **The Figure R1 has been shown in supplementary information as Figure S13. The Figure R2 has been shown in supplementary information as Figure S14. The Figure R3 has been shown in supplementary information as Figure S16. And the corresponding description has been shown in the revised manuscript on Page 7, lines 11–19 (highlight in yellow).**

Figure R1. (a) HRTEM and (b, c) HAADF-STEM images of the 10Ni/TiO₂-Ref1 catalyst. (d) EDS-mapping results of the 10Ni/TiO₂-Ref1 catalyst.

Figure R2. (a) HETEM images and (b) statistical analysis of particle size for Ni particles in Ni/TiO₂-Ref1 after the RWGS reaction.

Figure R3. (a, b) EELS-mapping results of the used 10Ni/TiO₂-Ref1 catalyst.

2. The relationship between SMSI and catalytic performances.

The large particles in Ni/TiO₂-Ref1 with an average size of ~27.6 nm exhibited very poor activity and did not have strong ability to generate the byproduct CH₄ (Figure R4). Therefore, it could be speculated that large particles of tens to hundreds of nanometers on Ni/TiO₂-OH did not possess the sufficient activity to catalyze the CO₂ hydrogenation and generate additional CH₄ (Figure R5). Although they were coated with the TiO_x layer, this SMSI was almost not related to the enhancement for the catalytic performance. **The catalytic performance of Ni/TiO₂-Ref1 has been added in Figure 4a with the corresponding description on Page 10, lines 5, 6 in the revised manuscript. Figure**

R5 has been shown in supplementary information as Figure S9 with corresponding description on Page 7, line 1 in the revised manuscript (highlight in yellow).

Figure R4. CO₂ conversion and CO selectivity for 10Ni/TiO₂-Ref1 under the GHSV of 400,000 mL·g_{cat}⁻¹·h⁻¹.

Figure R5. (a) HAADF-STEM images of the used 10Ni/TiO₂-OH used catalyst. (b) Statistical histogram of the size distribution of Ni particles for the used 10Ni/TiO₂-OH catalyst. (c, d) HAADF-STEM images for Ni particles of the used 10Ni/TiO₂-OH catalyst.

3. The relationship between size effects and catalytic performances (activity and

selectivity).

After the H₂ pretreatment and the following RWGS reaction, Ni species were stabilized on Ni/TiO₂-OH dominantly as Ni clusters with ~1 nm (Figure R6a, d), while in Ni/TiO₂-Ref1, Ni species were agglomerated into large particles of ~27.6 nm (Figure R6b, e). In order to increase the gradient of particle size distribution, we further reduced the temperatures of air calcination and H₂ pretreatment of Ni/TiO₂-Ref1 from 600 °C to 500 °C (the obtained sample was denoted as “Ni/TiO₂-Ref1-500”). TEM images results showed that the size of Ni species on the Ni/TiO₂-Ref1-500 catalyst after H₂ pretreatment at 500 °C was ~12.4 nm (Figure R6c, f). As illustrated in Figure R7a, The CO₂ conversion ranked in the order of Ni/TiO₂-OH > Ni/TiO₂-Ref1-500 > Ni/TiO₂-Ref1, which was negatively correlated with the ordering of the particle size. In order to adjust the CO selectivity for these three samples, we varied the gas hourly space velocity (GHSV) to regulate the comparability of CO₂ conversion (Figure R7b). The CO selectivity of Ni/TiO₂-Ref1-500 with Ni particle size of ~12.4 nm was only 21.3%, much lower than that of Ni/TiO₂-OH with Ni cluster size of ~1.0 nm. This result was consistent with the notion that large-size metal particle tends to be more active for methanation (Christopher, P. et al. *J. Am. Chem. Soc.* **2015**, 137, 3076–3084, Guo, L. M. et al. *ACS Catal.* **2023**, 13, 10364–10374). However, as the size increased to ~27.6 nm, even reducing the GHSV to 20,000 mL·g_{cat}⁻¹·h⁻¹, the CO₂ conversion still could not be adjusted to a comparable level with that of Ni/TiO₂-OH, indicating that the very poor activity of Ni/TiO₂-Ref1 with the meaningless high CO selectivity.

Based on above results, the relationship between the Ni size on TiO₂ and the catalytic performance in atmospheric CO₂ hydrogenation could be clearly revealed (Figure R8). **On the one hand, compared to highly dispersed Ni clusters, Ni particles induced the formation of CH₄. In addition, when the Ni particle size was larger than ~27 nm, it was meaningless to discuss the product selectivity due to the very low activity. The results clearly confirm the referee’s concept that the high CO selectivity at high CO₂ conversion stems from the size effect. The role of TiO₂-OH is to stabilize the 1 nm Ni clusters.**

The Figure R6a has been shown in revised manuscript as Figure 2b. The Figure R6d has been shown in supplementary information as Figure S8c. The Figure R6b, e has been shown in supplementary information as Figure S22. The Figure R6c, f has been shown in supplementary information as Figure S14. Figure R7a has been shown in supplementary information as Figure S23a. Figure R7b has been shown in revised manuscript as Figure 4b. Figure R8 has been shown in revised manuscript as Figure 4c. The corresponding description has been added on Page 10, lines 9–28 and Page 11, line 1 in the revised manuscript (highlight in yellow).

Figure R6. (a) HAADF-STEM images of Ni/TiO₂-OH after H₂ pretreatment and RWGS at 600 °C. (b) TEM images of Ni/TiO₂-Ref1 after H₂ pretreatment at 500 °C. (c) TEM images of Ni/TiO₂-Ref1 after H₂ pretreatment and RWGS at 600 °C. (d–f) The statistical analysis of particle size for Ni particles in (a–c), respectively.

Figure R7. (a) The CO₂ conversion and CO selectivity for 10Ni/TiO₂-OH and 10Ni/TiO₂-Ref1 after calcination and H₂ pretreatment at 600 °C, as well as 10Ni/TiO₂-Ref1 after calcination and H₂ pretreatment at 500 °C (10Ni/TiO₂-Ref1-500) under the GHSV of 400,000 mL·g_{cat}⁻¹·h⁻¹. (b) The CO₂ conversion and CO selectivity for three sample in (a) at 500 °C under various gas hourly space velocity.

Figure R8. The schema of the influence of Ni species size on catalytic performances for Ni/TiO₂ catalysts.

Comment 2: The DFT part seems a bit off-topic too, seemingly discussing the stability of the "single atom" Ni system.

Response: Reviewer's valuable comments are highly appreciated by us. During the process of revising the manuscript, we confirmed that the enhancement of catalytic activity could be attributed to the formed dense and stable ultra-small Ni clusters on 10Ni/TiO₂-OH rather than the SMSI. The formation of stable single Ni atoms on the hydroxylated TiO₂ was the prerequisite for constructing Ni clusters during the RWGS reaction. Therefore, it was necessary for us to perform the DFT calculation to explore reasons for the stabilization of isolated Ni atom on hydroxylated TiO₂. DFT calculations indicated that both electrostatic and covalent interactions within the Ni-O bonds contributed to the excellent stability of the adsorbed Ni atoms in the Ni/OH_{5c}-TiO₂-I and Ni/OH_{5c}-TiO₂-II models. During the subsequent H₂ pretreatment and RWGS reaction, Ni single atom primarily transformed into clusters and the potential mechanism of this transformation process was also explored by DFT simulations. The removal of the -OH group and the promotion of the Ni diffusion under the hydrogen atmosphere well explained the Ni agglomeration into clusters observed after catalysis. In addition, we also carried out the DFT simulations to explore the catalytic behavior that happened on the Ni cluster/TiO₂ catalyst. As shown in Figure R9, CO₂ and H₂ could be efficiently activated at the Ni cluster-TiO₂ interface, following by the formation of the active intermediates to promote the RWGS reaction via the optimal carboxyl pathway. **Figure R9 has been moved from the supplementary information to the revised manuscript as Figure 6.** Thanks for the reviewer's comment again.

Figure R9. (a) CO₂ adsorption on Ni₈/TiO₂ system and H₂ adsorption on CO₂-Ni₈/TiO₂ system. (b) Potential energy diagram for the RWGS reaction on Ni₈/TiO₂(101) via carboxyl and formate pathways. The illustration show the structures of some intermediate and transition states. The complete reaction pathways are shown in Supplementary Fig. 37 (carboxyl) and S38 (formate). “TS” represents transition state.

Comment 3: *Figure 5 is the least convincing, and I believe the scheme in Figure 5d-e is not backed by evidence.*

Response: Thanks for reviewer’s valuable comments. Further experimental investigations were conducted to explore the reasons for the encapsulation of Ni particles. As we all know, large metal particles were prone to be encapsulated by the reducible oxide support. However, in our work, Ni particles were encapsulated by TiO₂-OH but not by TiO₂-Ref1, indicating that the encapsulation of large metal particles was not a universal phenomenon. H₂-TPR experiments (Figure R10a) and the corresponding actual hydrogen consumption (Figure R10b) showed that the actual hydrogen consumption of Ni/TiO₂-OH catalyst was more than the theoretical hydrogen

consumption of NiO, suggesting that more surface oxygen atoms of TiO₂-OH support were reduced under H₂ flow. While the actual hydrogen consumption of Ni/TiO₂-Ref1 was close to the theoretical hydrogen consumption, confirming that the weak MSI between Ni species and TiO₂-Ref1 (Figure R10b). The limited reducibility of TiO₂-Ref1 and the weak MSI explained why large Ni particles could not be coated by TiO₂ support over 10Ni/TiO₂-Ref1.

Therefore, the disparity in encapsulation phenomena of large-size Ni particles over these two catalysts arose from the reducibility of TiO₂ supports and the metal-support interaction, rather than the surface energy difference. Once again, we sincerely thank the reviewer for the valuable suggestions. **The Figure R10a has been shown in revised manuscript as Figure 5a. The Figure R10b has been shown in supporting information as Figure S28. The corresponding description has been added on Page 13, lines 19–27 in the revised manuscript (highlight in yellow).**

Figure R10. (a) H₂-TPR profiles of the NiO, 10Ni/TiO₂-Ref1 and 10Ni/TiO₂-OH samples. (b) H₂ consumption calculation from H₂-TPR of Ni/TiO₂-OH and Ni/TiO₂-Ref in (a).

As a result, based on reviewer's valuable comments, **we have carefully revised the title, abstract and introduction of this work and we believe that the current version will deepen the previous understanding of CO₂ hydrogenation over Ni-based catalysts, cluster catalysts construction and the metal-support interaction.** The highlights of this work can be divided into the following points:

(1) **Constructing the sintering-resistant Ni cluster catalyst.** Hydroxylated TiO₂ with abundant –OH groups disperse Ni species into isolated Ni atoms, which induces the formation of dense and stable Ni clusters under the RWGS reaction with harsh conditions.

(2) **Breaking the activity-selectivity trade-off of CO₂ hydrogenation at atmospheric pressure.** The Ni catalysts clearly shows the size dependent RWGS performance. Due to the Ni cluster/TiO₂ interfaces with enhanced CO₂ activation and weak CO adsorption, the Ni cluster/TiO₂ catalyst achieves the combination of high activity, high CO selectivity and excellent stability, which exceeds almost all reported catalysts.

References

1. Matsubu, J. C. et al. Isolated Metal Active Site Concentration and Stability Control Catalytic CO₂ Reduction Selectivity. *J. Am. Chem. Soc.* **137**, 3076–3084 (2015).
2. Yang, B. et al. Size-Dependent Active Site and Its Catalytic Mechanism for CO₂ Hydrogenation Reactivity and Selectivity over Re/TiO₂. *ACS Catal.* **13**, 10364-10374 (2023).

Reviewer #2: Authors report the frustration of SMSI in the Ni/TiO₂ system by pre-functionalizing the support with OH before Ni deposition. The designed catalyst shows higher conversion than the catalyst without -OH for CO₂ hydrogenation. Further, CO selectivity of Ni/TiO₂-OH was higher than using other supports (Al₂O₃, CeO₂). The work is interesting, both major revisions are needed before it can be reconsidered for acceptance.

Response: The reviewer's comments and suggestions are of great help to improve our research work. Based on reviewer's comments, we have carefully revised the manuscript.

Comment 1: Fig.2 is missing, making it difficult to assess the interpretation of data.

Response: Thanks for reviewer's comment. Fig. 2 has been attached to our manuscript as Fig. R11, which illustrated the structural information of the used 10Ni/TiO₂-OH catalyst after the RWGS reaction. Thanks for reviewer's comments again.

Fig. R11. Structural information of the used 10Ni/TiO₂-OH catalyst. (a) Schematic illustration of the generation process for the used 10Ni/TiO₂-OH catalysts from the as-

prepared 10Ni/TiO₂-OH catalyst. (b, c) HAADF-STEM images of the used 10Ni/TiO₂-OH catalyst. The EDS-elemental mapping results for the Ni cluster region (d) and Ni particle region (e) over the used 10Ni/TiO₂-OH catalyst.

Comment 2: The callout of Figures in SI is messy and makes the reader run through pages to follow the story.

Response: Thanks for reviewer's valuable comments and suggestions. According to reviewers' comments, we have carefully revised our manuscript, including the language, the discussion and analysis of data. Thanks for reviewer's comments again.

Comment 3: Particle size distributions are needed to assess the precision of the synthesis methods.

Response: Thanks for reviewer's valuable comments and suggestions. As illustrated in EXAFS profiles (Figure R12a), the absence of the Ni-Ni coordination for the fresh 10Ni/TiO₂-OH catalyst indicated that before the reaction, Ni species were mainly dispersed as isolated Ni atoms on the TiO₂-OH support. For the used 10Ni/TiO₂-OH catalyst, the statistical histogram of size distribution over Ni clusters (particle number = 100) was shown in Figure R12b. **The average size of Ni clusters was about 1.0 nm, and Ni clusters were uniform in size. In addition, the average size of the Ni particles over the fresh and used 10Ni/TiO₂-Ref1 was about 9.6 nm and 27.6 nm, respectively (Figure R12c, d).** Therefore, the synthesis method in this work provided a facile pathway to obtain Ni cluster catalysts. **The Figure R12a has been shown in the supplementary information as Figure S6b. The Figure R12b has been shown in the supplementary information as Figure S8c. The Figure R12c has been shown in the supplementary information as Figure S13c. The Figure R11d has been shown in the supplementary information as Figure S14b.**

Fig R12. (a) Extended X-ray absorption fine structure (EXAFS) profiles. (b) The statistical histogram of size distribution of Ni clusters (cluster number = 100) for the used 10Ni/TiO₂-OH catalyst. (c, d) The statistical histogram of size distribution of the Ni particle (particle number = 100) for the fresh (c) and used (d) 10Ni/TiO₂-Ref1 catalyst.

Comment 4: The existence/non-existence of SMSI relies heavily on localized STEM images, which can be misleading. Authors need to corroborate the existence/non-existence of SMSI via CO or N₂O chemisorption and/or DRIFTS, as well as compare the density of sites for the hydroxylated and non-hydroxylated sample before and after reaction.

Response: The reviewer's comment is highly appreciated by us. The CO adsorption at 30 °C of two samples after the RWGS reaction by *in situ* DRIFTS was carried out (Figure R13). Both samples exhibited the similar results, with only the gaseous CO

signal observed and no signal attributing CO adsorbed on Ni species, which suggested that the Ni species in both samples had a low CO adsorption capacity. As the temperature was lower to $-20\text{ }^{\circ}\text{C}$, the bands that appeared at 1850 cm^{-1} and 1905 cm^{-1} in Ni/TiO₂-OH and Ni/TiO₂-Ref1, respectively, could be attributed to 3-fold bound carbonyls on Ni⁰ sites. To further confirm the presence of Ni clusters observed from the HAADF-STEM images of 10Ni/TiO₂-OH, the *in situ* infrared spectroscopy at low temperature (130 K) by using CO as a probe molecule after RWGS reaction was conducted (Figure R14). Three bands at ~ 2170 , 2152 , and 2093 cm^{-1} were observed. The former was attributed to CO adsorption on Ti⁴⁺ sites. The bands of 2152 cm^{-1} and 2093 cm^{-1} were assigned to CO linearly adsorbed on Ni²⁺ and Ni⁰ sites, respectively. The coexistence of these two bands proved the presence of Ni clusters after the reaction.

These results above indicated the presence of unencapsulated Ni species in both samples. Combined with the findings from HAADF-STEM images (Figure R15, 16), it was observed that **the large Ni particle did not undergo SMSI with the TiO₂-Ref1, while did with the TiO₂-OH. Only Ni clusters exposed as active sites in Ni/TiO₂-OH sample.**

The Figure R14 has been shown in the revised manuscript as Figure 5c with the corresponding description on Page 15, lines 7–15 (highlight in yellow). The Figure R15 has been shown in the supplementary information as Figure S16 with the corresponding description on Page 7, lines 17–19 (highlight in yellow). The Figure R16 has been shown in the supplementary information as Figure S11 with the corresponding description on Page 7, lines 3–5 (highlight in yellow).

Figure R13. *In situ* diffuse reflectance infrared Fourier transform (DRIFTS) spectroscopy after exposing the Ni/TiO₂ catalyst to CO at 30 °C (a) and -20 °C (b) after the RWGS reaction under 600 °C.

Figure R14. *In-situ* infrared spectra recorded after exposing the 10Ni/TiO₂-OH catalysts to CO with different partial pressure at 130 K after the RWGS reaction.

Figure R15. (a, b) The EELS-mapping results of the used 10Ni/TiO₂-Ref1 catalyst.

Figure R16. (a–b) The EELS-mapping results for Ni particles of the used 10Ni/TiO₂-OH catalyst.

Comment 5: Authors claim that 10Ni/TiO₂-Ref1 is underperforming compared to 10Ni/TiO₂-OH catalyst because of SMSI; however, 10Ni/TiO₂-Ref1 has much larger Ni particles and thus less sites exposed, therefore lower conversion is not surprising. Authors need to compare performances on the basis of exposed Ni sites.

Response: Reviewer's valuable comments and suggestions are highly appreciated. Based on HAADF-STEM image of the 10Ni/TiO₂-Ref1 catalyst (Figure R15), no TiO_x

layer was found on the surface of Ni particles. Therefore, Ni clusters on Ni/TiO₂-OH and large Ni particles on Ni/TiO₂-Ref1 were exposed. According to the literature reported (Kwak, J. H. et al. *ACS Catal.* **2021**, 11, 5894–5905; Deng, D. et al. *Nat. Catal.* **2023**, 6, 1052–1061), the relative area of the CO adsorption peak is positively correlated with quantified amount of exposed metal sites. By analyzing the CO adsorption peaks on Ni sites detected by *in-situ* DRIFTS (Figure R13b), it was found that the amount of exposed Ni sites in Ni/TiO₂-OH was 3.5 times higher than in Ni/TiO₂-Ref1. As a result, the CO₂ conversions per unit Ni sites at 500 °C were 13.8% (Ni/TiO₂-OH) and 3.4% (Ni/TiO₂-Ref1), respectively. This indicated that the cluster sites in Ni/TiO₂-OH exhibited superior catalytic activity compared to the particle sites in Ni/TiO₂-Ref1. To investigate the reasons for the difference in catalytic property between distinct structures, CO₂-TPD experiments were carried out. As illustrated as Figure R17, no obvious signal was observed in the CO₂-TPD for the Ni/TiO₂-Ref1 catalyst, indicating that the severe sintering of the large Ni particles resulted in the loss of active sites for CO₂ adsorption, leading to extremely poor activity. Compared to 10Ni/TiO₂-Ref1 with weak capacity to adsorb CO₂, two strong CO₂ desorption peaks appeared over 10Ni/TiO₂-OH. By comparing the result of Ar-TPD, the first feature at low temperature could be ascribed to chemisorbed CO₂ molecules, and the later one was related to the decomposition of surface carbides. The strong CO₂ desorption signal indicated that compared to 10Ni/TiO₂-Ref1, rich Ni cluster-TiO₂ interfaces might be served as crucial sites for absorbing CO₂. Therefore, **the significant difference in activity between these two catalysts originated from the size effect of the exposed Ni species.** Ni clusters anchored on the TiO₂-OH support, which could remain stable under high-temperature reducing conditions, were crucial for catalyzing the RWGS reaction efficiently. **The Figure R17 has been shown in the supplementary information as Figure S33 with the corresponding description on Page 16, lines 8–16 (highlight in yellow).**

Fig R17. CO₂-TPD profiles for the 10Ni/TiO₂-Ref1 and 10Ni/TiO₂-OH catalysts and Ar-TPD profile for the 10Ni/TiO₂-OH catalysts.

Comment 6: For 10Ni/TiO₂-Ref1, were Ti³⁺ species were observed in XPS? thus supporting SMSI.

Response: Thanks for reviewer's valuable comments and suggestions. Based on the reviewer's comment, we further explore the structure of the Ni/TiO₂-Ref1 catalyst via HAADF-STEM images. As shown in Figure R15, no TiO₂ layer was found on the surface of Ni particles, implying SMSI was not occurred in Ni/TiO₂-Ref1 catalyst. In addition, X-ray Photoelectron Spectroscopy (XPS) characterization of 10Ni/TiO₂-Ref1 was also conducted (Figure R18). The peak at 458.5 eV was identified as Ti⁴⁺. There was no shift observed in the Ni/TiO₂-Ref1 catalyst before and after the reaction, indicating the absence of Ti³⁺ species. Moreover, only Ti⁴⁺ species but no obvious Ti³⁺ signal could also be found for 10Ni/TiO₂-OH. We speculated that even though the MSI between Ni and TiO₂-OH facilitated the partial reduction of TiO₂-OH, the limited concentration of surface Ti³⁺ sites were below the detection line of XPS techniques. According to previous report, the absence of surface oxygen vacancies might be caused by the easy mobility of oxygen vacancies from surface to bulk on anatase TiO₂ (Scheiber, P. et al. *Phys. Rev. Lett.* **2012**, 109, 136103). **The Figure R18 has been**

shown in the supplementary information as Figure S32a with the corresponding description on Page 15, lines 23–28 and Page 16, lines 1, 2 (highlight in yellow).

Figure R18. XPS spectra and corresponding fitting curves of Ti $2p$ in the 10Ni/TiO₂-Ref1 and 10Ni/TiO₂-OH.

References

1. Kwak, J. H. *et al.* CH₄ oxidation activity in Pd and Pt–Pd bimetallic catalysts: correlation with surface PdO_x quantified from the DRIFTS study. *ACS Catal.* **11**, 5894–5905 (2021).
2. Deng, D. *et al.* Direct conversion of methane with O₂ at room temperature over edge-rich MoS₂. *Nat. Catal.* **6**, 1052–1061 (2023).
3. Scheiber, P. *et al.* Surface mobility of oxygen vacancies at the TiO₂ anatase (101) surface. *Phys. Rev. Lett.* **109**, 1–5 (2012).

Reviewer #3: *In this manuscript the authors report the catalytic performance of CO₂ reduction to CO by 1 nm Ni clusters supported on hydroxylated TiO₂. They ascribe the CO selectivity to the inhibited SMSI effect. The formation mechanism of SMSI effect is indeed a critical but also challenging issue in heterogeneous catalysis. This is a good fundamental point to discuss, but there are some issues that should clarify. The language of the manuscript is not professional, and should be polished thoroughly.*

Response: Reviewer's valuable comments and suggestions are highly appreciated by us. Based on your comments, we have revised the manuscript, and the language of the manuscript has been polished. We determined that the much improved activity of Ni/TiO₂-OH was attributed to the size effect of Ni species, rather than the SMSI-induced TiO_x overlayer on Ni particles. **We believe that the current version will deepen the previous understanding of CO₂ hydrogenation over Ni-based catalysts, cluster catalysts construction and the metal-support interaction.** At present, the highlights of this work can be summarized into the following points:

(1) Constructing the sintering-resistant Ni cluster catalyst. Hydroxylated TiO₂ with abundant -OH groups dispersed Ni species into isolated Ni atoms, which induced the formation of dense and stable Ni clusters under the RWGS reaction with harsh conditions.

(2) Breaking the activity-selectivity trade-off of CO₂ hydrogenation at atmospheric pressure. Due to the Ni cluster/TiO₂ interfaces with enhanced CO₂ activation and weak CO adsorption, the Ni cluster/TiO₂ catalyst achieves the combination of high activity, high CO selectivity and excellent stability, which exceeds almost all reported catalysts.

Comment 1: *For CO₂ hydrogenation, Ru and Ni are catalysts for CH₄ production; however, SMSI, decreasing particle size of metals, and increasing temperature will all increase CO selectivity. In this work the reaction temperature is 500 °C, how to exclude the effects of TiO_x overlayers. How to consider the effects of particle size and temperature?*

Response: Thanks for the reviewer's valuable comments. In order to respond to reviewer's comments, we have done additional experiments and analyzed the results as below:

1. The effects of TiO_x overlayers.

In our manuscript, we identified the coverage of large-sized Ni particles on the 10Ni/TiO₂-OH catalyst induced by SMSI. We further explore the structure of the 10Ni/TiO₂-Ref1 catalyst via HAADF-STEM images. As shown in Figure R19, no TiO_x layer was found on the surface of Ni particles. However, despite the exposure of Ni particles could act as active sites, the Ni/TiO₂-Ref1 catalyst still catalyzed the generation of CO instead of CH₄ (Figure R20). This was due to the sintering of Ni particles (~27 nm) on TiO₂-Ref1, resulting in the loss of sufficient sites to adsorb CO molecules, as evidenced by *in situ* DRIFTS of CO adsorption experiment at 30 °C after the RWGS reaction (Figure R21). Since the large particles in Ni/TiO₂-OH were also several tens to hundreds of nanometers in size, it could be inferred that these large particles were also inactive and did not generate additional CH₄ (Figure R22). Therefore, although they were coated with the TiO_x layer, this layer was almost not related to the enhancement of the catalytic performance. **Figure R19 has been shown in the supplementary information as Figure S16 with corresponding description on Page 7, lines 3–5 in the revised manuscript. The catalytic performance of Ni/TiO₂-Ref1 in Figure R20 has been added in Fig.4 with the corresponding description on Page 10, lines 5, 6 in the revised manuscript. Figure R22 has been shown in the supplementary information as Figure S9 with the corresponding description on Page 7, line 1 in the revised manuscript (highlight in yellow).**

Figure R19. (a, b) The EELS-mapping results of the used 10Ni/TiO₂-Ref1 catalyst.

Figure R20. The CO₂ conversion and CO selectivity for 10Ni/TiO₂-Ref1 under the GHSV of 400,000 mL·g_{cat}⁻¹·h⁻¹.

Figure R21. *In situ* diffuse reflectance infrared Fourier transform (DRIFTS) spectroscopy after exposing the Ni/TiO₂ catalyst to CO at 30 °C.

Figure R22. (a) HAADF-STEM images of the used 10Ni/TiO₂-OH catalysts. (b) Statistical histogram of size distribution of the Ni particles for the used 10Ni/TiO₂-OH catalyst. (c, d) HAADF-STEM images for Ni particles of the used 10Ni/TiO₂-OH catalysts.

2. The effect of particle size.

After the H₂ pretreatment and the RWGS reaction, the size of Ni clusters in Ni/TiO₂-OH was ~1 nm (Fig. R23a, d), while Ni species in the used Ni/TiO₂-Ref1 catalyst sintered to ~27.6 nm (Figure R23b, e). In order to increase the gradient of particle size distribution, we reduced the air calcination and H₂ pretreatment of Ni/TiO₂-Ref1 from 600 °C to 500 °C (denoted as “Ni/TiO₂-Ref1-500”). TEM images showed that the size of Ni species was ~12.4 nm (Figure R23c, f). As illustrated in Figure R24a, The CO₂ conversion ranked in the order of Ni/TiO₂-OH > Ni/TiO₂-Ref1-500 > Ni/TiO₂-Ref1, which was opposite to the ordering of particle sizes. In order to compare the CO selectivity of these three samples in a more rational way, we varied the gas hourly space velocity (GHSV) to regulate the comparability of CO₂ conversion (Figure R24b). The CO selectivity of Ni/TiO₂-Ref1-500 with Ni particle size of 12.4 nm was only 21.3%, much lower than that of Ni/TiO₂-OH with Ni cluster size of 1.0 nm, which was consistent with the notion that larger metal particles tend to be more active for methanation (Christopher, P. et al. *J. Am. Chem. Soc.* **2015**, 137, 3076–3084, Guo, L. M. et al. *ACS Catal.* **2023**, 13, 10364–10374). However, as the Ni size increased to 27.6 nm, the CO₂ conversion could not be tuned to a comparable level even when the GHSV was lowered to 20,000 mL·g_{cat}⁻¹·h⁻¹, suggesting that the activity of Ni/TiO₂-Ref1 was very poor and the CO selectivity was meaningless high.

Based on above results, the relationship between the Ni size on TiO₂ and the catalytic performance in atmospheric CO₂ hydrogenation could be clearly revealed (Figure R25). **On the one hand, compared to highly dispersed Ni clusters, Ni particles induced the formation of CH₄ and the reduced CO₂ conversion. On the other hand, when the Ni particle size was larger than 27 nm, it was meaningless to discuss the product selectivity due to the very low activity.**

The Figure R23a has been shown in revised manuscript as Figure 2b. The Figure R23d has been shown in supplementary information as Figure S8c. The Figure R23b, e has been shown in supplementary information as Figure S22. The Figure R23c, f has been shown in supplementary information as Figure S14. Figure R24a

has been shown in supplementary information as Figure S23a. Figure R24b has been shown in revised manuscript as Figure 4b. Figure R25 has been shown in revised manuscript as Figure 4c. The corresponding description has been added on Page 10, lines 9–28 and Page 11, line 1 in the revised manuscript (highlight in yellow).

Figure R23. (a) HAADF-STEM images of Ni/TiO₂-OH after H₂ pretreatment and RWGS at 600 °C and (b, c) TEM images of Ni/TiO₂-Ref1 after H₂ pretreatment and RWGS at 500 °C (b) and 600 °C (c). (d–f) The statistical analysis of particle size for Ni particles in (a–c), respectively.

Figure R24. (a) The CO₂ conversion and CO selectivity for 10Ni/TiO₂-OH, 10Ni/TiO₂-Ref1 (air calcination and H₂ pretreatment at 600 °C) and 10Ni/TiO₂-Ref1-500 (air calcination and H₂ pretreatment at 500 °C) under the GHSV of 400,000 mL·g_{cat}⁻¹·h⁻¹. (b) The CO₂ conversion and CO selectivity for these three samples at 500 °C under different GHSVs.

Figure R25. The schema of the influence of Ni species size on the catalytic performance over Ni/TiO₂ catalysts.

3. The effect of the reaction temperature.

For the atmospheric CO₂ hydrogenation driven by thermal energy, CO and CH₄ are two main products. Thermodynamically, CH₄ is the major product for CO₂ hydrogenation at relative low operating temperatures (Figure R26). Obviously, low operating temperatures favor methanation, while high reaction temperatures favor the RWGS reaction. However, as shown in Figure R27, compared to reference 10Ni/Al₂O₃ and 10Ni/CeO₂ catalysts, the 10Ni/TiO₂-OH catalyst showed much higher CO selectivity from 400 °C to 600 °C, indicating that 10Ni/TiO₂-OH could efficiently catalyze the RWGS reaction rather than methanation. Therefore, the test temperature was not the critical factor to control the product selectivity for 10Ni/TiO₂-OH.

Figure R26. CO₂ equilibrium conversion for (a) RWGS reaction and (b) Methanation from 400 °C to 600 °C (CO₂:H₂ = 1:3).

Figure R27. The CO selectivity for the 10Ni/TiO₂-OH, 10Ni/CeO₂ and 10Ni/Al₂O₃ catalysts (GHSV = 400,000 mL·g_{cat}⁻¹·h⁻¹).

Comment 2: According to the catalytic condition, GHSV should be 400,000 mL·g_{cat}⁻¹·h⁻¹ not 800,000 mL·g_{cat}⁻¹·h⁻¹.

Response: Thanks for reviewer's valuable comments and suggestions. In this manuscript, we used a GHSV of 400,000 mL·g_{cat}⁻¹·h⁻¹ to evaluate the temperature-dependent activity. However, the CO₂ conversion of Ni/TiO₂-OH reached the thermodynamic limitation at 600 °C. Under such a GHSV, the stability of the catalyst could not be scientifically accessed. **In order to better track the deactivation behavior, we used a much higher GHSV (800,000 mL·g_{cat}⁻¹·h⁻¹) in the long-term stability test. Based on reviewer's comments, we have added more details to the test conditions on Page 19, lines 28, 29.** Thanks for reviewer's comments again.

Comment 3: *Figure caption of Fig. 4a is not correct.*

Response: Reviewer's comments and suggestions are highly appreciated. We have modified the content in Figure 4a, changing the comparison between Ni/TiO₂-OH and the reference catalysts of Ni/CeO₂ and Ni/Al₂O₃ to a comparison with Ni/TiO₂-Ref1. And the relevant Figure caption was corrected to "The CO₂ conversion and CO selectivity for 10Ni/TiO₂-OH and 10Ni/TiO₂-Ref1 catalysts under the GHSV of 400,000 mL·g_{cat}⁻¹·h⁻¹". Thanks for reviewer's comments again.

Comment 4: *The scheme of SMSI in Fig. 5e is not reasonable. As the mechanism of SMSI is quite complicated, the author should provide solid supports for their guess and conclusions.*

Response: Thanks for reviewer's valuable comments. Further experimental investigations were conducted to explore the reasons for the encapsulation of Ni particles. As we all know, large metal particles were prone to be encapsulated by the reducible oxide supports. However, in our work, Ni particles were encapsulated by TiO₂-OH but not by TiO₂-Ref1, indicating that the encapsulation of large metal particles was not a universal phenomenon. To investigate the disparity in the encapsulation phenomena, H₂-TPR experiments (Figure R28a) and the corresponding actual hydrogen consumption (Figure R28b) were shown. The actual hydrogen consumption of Ni/TiO₂-OH was higher than the theoretical hydrogen consumption calculated based on NiO, suggesting that more surface oxygen atoms of the TiO₂-OH support was involved in the reduction. Besides, the actual hydrogen consumption of Ni/TiO₂-Ref1 was close to the theoretical hydrogen consumption, confirming that the weak MSI between Ni species and TiO₂-Ref1 (Figure R28b), which resulted in the inability of the TiO₂-Ref support to cover even large Ni particles through the strong metal-support interaction (SMSI).

Therefore, the disparity in the encapsulation phenomenon of these two catalysts arose from the reducibilities of TiO₂ supports and the metal-support interaction, rather than the surface energy difference. The schematic diagrams in Fig. 5d–e were not reasonable.

Once again, thanks for the reviewer's valuable suggestions. **The Figure R28a has been shown in revised manuscript as Figure 5a. The Figure R28b has been shown in supplementary information as Figure S28. The corresponding description has been added on Page 13, lines 19–27 in the revised manuscript (highlight in yellow).**

Figure R28. (a) H₂-TPR profiles of the NiO, 10Ni/TiO₂-Ref1 and 10Ni/TiO₂-OH samples. (b) H₂ consumption calculated based on H₂-TPR of Ni/TiO₂-OH and Ni/TiO₂-Ref in (a).

Comment 5: Line 50: should be driving force; what is meaning of the very Ni large particle in line 114.

Response: Thanks for reviewer's valuable comments. According to reviewer's comments, we have revised the discussion of the SMSI. As shown in Figure R29, after the activity evaluation, isolated Ni atoms on the TiO₂-OH support transformed into abundant Ni clusters and a few large-sized Ni particles. And large-sized Ni particles were encapsulated by TiO_x support induced by the SMSI (Figure R30). Thanks for reviewer's comments again. **The Figure R29 has been shown in revised manuscript as Figure 2b, c. The Figure R30 has been shown in supplementary information as Figure S11. The corresponding description was shown on Page 6, lines 14, 15 and Page 7, lines 1 in the revised manuscript (highlight in yellow).**

Figure R29. (a, b) The HAADF-STEM image of used 10Ni/TiO₂-OH used catalysts.

Figure R30. (a, b) EELS-mapping results for Ni particles of the used 10Ni/TiO₂-OH catalyst.

References

1. Matsubu, J. C. et al. Isolated Metal Active Site Concentration and Stability Control Catalytic CO₂ Reduction Selectivity. *J. Am. Chem. Soc.* **137**, 3076–3084 (2015).
2. Yang, B. et al. Size-Dependent Active Site and Its Catalytic Mechanism for CO₂ Hydrogenation Reactivity and Selectivity over Re/TiO₂. *ACS Catal.* **13**, 10364-10374 (2023).

Reviewer 4#: *The authors aimed to address the challenge of finding efficient and cost-effective catalysts for the reverse water gas shift (RWGS) reaction, crucial for CO₂ hydrogenation. Although Ni nanoparticles are excellent catalysts for hydrogenation, they are prone to encapsulation through classical strong metal-support interaction (SMSI) with sub-oxide species at high reduction temperatures, potentially limiting their use in the RWGS reaction. To overcome this, the authors claim to create SMSI-resistant Ni-clusters on hydroxylated TiO₂. The authors claim to deposit Ni in single-atom form on the surface and then transforming them into clusters after hydrogen treatment, thereby they avoid the coating or encapsulation of Ni by TiO_{2-x} also known as the SMSI effect. This approach enabled enhanced catalytic performance and stability at temperatures as high as 600°C for the RWGS reaction.*

While I do believe the authors have interesting work here, I have a few conceptual issues with the work. Firstly, it is well-known that encapsulation by SMSI occurs less on nanoclusters than on nanoparticles due to the concentration difference of phases. I.e., typically, due to the concentration difference which causes phase-diffusion of sub-oxide species, metal clusters are actually much less susceptible to encapsulation by “reducible” MOs. So the authors claim to have come up with a new way to avoid something, which in fact was already well-known. Secondly, the authors fail to properly characterize their TiO₂-OH support. TiO₂ itself is active towards RWGS, and it may very well be that the functionalization created a change in the electronic/geometric structure of the support itself which is responsible for the observed changes, which should have been assessed; the electronic structure of the support post-treatment, and as well proper catalytic blank experiments should have been performed. I.e., proof is lacking where the additional activity/stability actually comes from. It could be a particle size effect of the metal, an electronic/geometric effect of the support itself, the availability of different sites, etc. Secondly, I have issues with the nomenclature; “SMSI-resistant”. MSI is a very broad phenomenon and I guess the authors are only referring to encapsulation/sub-oxide species covering? They should specifically discuss this

aspect then. Indeed of course clusters by definition have a much higher MSI effect, and actually probably that is what is causing the high activity, the fact that the nanoclusters indeed very much feel the metal support interaction therefor this terminology is at its best confusing and at its worst incorrect. Thirdly, the authors claim that SMSI should by default be avoided, and miss relevant literature showing that this is actually not always the case.

All in all, I believe that the hydroxylation approach gave the authors interesting results, but the novelty of the findings is arguable (both from the encapsulation side, and from the approach side). Furthermore, the observed effects are not properly characterized/corroborated. Thus, I am of the opinion that this manuscript is better suited for a different journal.

Response: Reviewer's valuable comments and suggestions are highly appreciated by us. Based on your comments, we have revised the whole manuscript carefully. We determined that the much improved activity of Ni/TiO₂-OH was attributed to the size effect of Ni species, rather than the SMSI-induced TiO_x overlayer on Ni particles. At present, the highlights of this work can be summarized into the following points:

(1) Converting commercial TiO₂ into H₂Ti₃O₇ with –OH groups. Through the hydrothermal treatment under an alkaline environment and the subsequent washing with HCl solution, commercial TiO₂ was converted into H₂Ti₃O₇ with high specific surface area and intrinsic –OH groups. Compared to commercial TiO₂ particles, the tubular H₂Ti₃O₇ can serve as an ideal support to disperse Ni species via the interaction between Ni and –OH.

(2) Constructing the sintering-resistant Ni cluster catalyst. Hydroxylated TiO₂ with abundant –OH groups dispersed Ni species into isolated Ni atoms, which induced the formation of dense and stable Ni clusters under the RWGS reaction with harsh conditions.

(3) Breaking the activity-selectivity trade-off of CO₂ hydrogenation at atmospheric pressure. Due to the Ni cluster/TiO₂ interfaces with enhanced CO₂ activation and weak CO adsorption, the Ni cluster/TiO₂ catalyst achieves the combination of high activity, high CO selectivity and excellent stability, which exceeds almost all reported catalysts.

Comment 1: “The Ni atoms were deposited 78 on the hydroxylated TiO₂ in single-atom form and subsequently transformed to clusters after hydrogen 79 treatment” “Ni 94 atomically anchored on the TiO₂-OH supports via a further calcination at 600 °C (denoted by 95 10Ni/TiO₂-OH, Fig. 1d).”

Proof is lacking to claim “atomic” sites, or single atoms. These are extremely difficult to properly characterize, and a few HAADF-STEM images are not enough to claim this, and X-ray EDS should actually even not show atomically dispersed sites; but rather suggests clusters? Also, it is said that XRD and XAFS show co-presence; but how can a technique it prove co-presence if both are bulk/ensemble average techniques?

Response: Thanks for reviewer’s valuable comments. From HAADF-STEM images (Figure R31), we could not find any Ni cluster and Ni particle, which suggested that Ni species was highly dispersed on the TiO₂-OH support before the reaction. This result was also verified by the EDS mapping. However, as the comment provided by the reviewer, the absence of Ni cluster and particle in HAADF-STEM images could not confirm that all loaded Ni was anchored on the support in the form of isolated atoms. The discussion of HAADF-STEM and X-ray EDS has been changed to “**The high dispersion of Ni over 10Ni/TiO₂-OH was confirmed**” on Page 5, lines 14, 15 in the revised manuscript.

For the EXAFS result, the Ni-Ni coordination was not detected (Figure R32). Although EXAFS was a bulk/ensemble average technique, the absence of Ni-Ni bond confirmed that **Ni species was dispersed on the support dominantly as single Ni atoms**. Based on reviewer’s comments, we have emphasized that “**Ni species was dispersed on the TiO₂-OH support dominantly as single Ni atoms, though the**

presence of a small number of sub-nanometer Ni clusters could not be ruled out” on Page 5, lines 18–20 and Page 6, line 1 in the revised manuscript.

Thanks for reviewer’s comments again. The Figure R31 has been shown in supplementary information as Figure S5. The Figure R32 has been shown in supplementary information as Figure S6b.

Figure R31. (a) HRTEM and (b, c) HAADF-STEM images of the 10Ni/TiO₂-OH catalyst. (d, e) EDS elemental mappings of the 10Ni/TiO₂-OH catalyst.

Figure R32. Extended X-ray absorption fine structure (EXAFS) profiles.

Comment 2: Also, characterization of the reaction mechanism/proving that extra activity comes from the stabilization (electronic effect of the support on the Ni clusters) and not from the defect sites in TiO₂ which itself is also active in RWGS?

Responses: We thank for the reviewer's comment. To investigate the effect of defect sites of TiO₂ in the RWGS reaction, we employed various characterization to confirm the low content of the surface vacancy. Firstly, as displayed by XPS Ti 2*p* spectra of as-prepared, used Ni/TiO₂-OH and Ni/TiO₂-Ref1, only the presence of Ti⁴⁺ was observed at 458.5 eV (Figure R33). Moreover, only Ti⁴⁺ was examined in the Ti L₃ edge near ambient pressure (NAP) – near edge X-ray absorption fine structure (NEXAFS) under H₂ and CO₂+H₂ conditions (Figure R34), consistent with the XPS results. Based on above results, we speculated that even though the MSI between Ni and TiO₂-OH facilitated the partial reduction of TiO₂, the limited concentration of surface Ti³⁺ sites were below the detection line of XPS and NEXAFS techniques. According to previous report, the absence of surface oxygen vacancies might be caused by the easy mobility of oxygen vacancies from surface to bulk on anatase TiO₂ (Scheiber, P. et al. *Phys. Rev. Lett.* **2012**, 109, 136103). Therefore, the contribution of surface oxygen vacancies to the catalytic activity enhancement was small. Thanks for reviewer's comments again. **The Figure R33, 34 has been shown in supplementary information as Figure S32 with the corresponding discussion on Page 15, lines 27, 28 and Page 16, lines 1, 2 in the revised manuscript (highlight in yellow).**

Figure R33. XPS spectra and corresponding fitting curves of Ti $2p$ in the 10Ni/TiO₂-OH and 10Ni/TiO₂-Ref1 catalysts before and after the reaction.

Figure R34. The *in situ* Ti L-edge NAP-NEXAFS profile of the 10Ni/TiO₂-OH catalyst.

Comment 3: “However, Ni 39 nanoparticles are very susceptible to strong metal-support interaction (SMSI) with reducible oxides, 40 causing the encapsulation of Ni particles at high temperatures with strong reductive atmosphere.11–13” -- Indeed, nanoparticles, but clusters much less. You state it yourself: “Larger metal nanoparticles with a higher surface energy are more inclined to be encapsulated, when the

minimization of surface free energy of metal nanoparticles is the major driven force. 16-19

Responses: The reviewer's comment is highly appreciated by us. To determine whether large-sized Ni nanoparticles are necessarily encapsulated by reducible oxides, we supplemented the corresponding HRTEM images, HAADF-STEM images and EDS-mapping results to investigate the structure of Ni/TiO₂-Ref1 catalyst. As shown in Figure R35, large-sized Ni particles could be observed in TiO₂-Ref1 support surface. And the average size of Ni particles was about 9.6 nm (the inset in Figure R35c). After the RWGS reaction at 600 °C, the Ni particles further aggregated into larger size of about 27.6 nm (Figure R36). The EELS-mapping results of two different Ni particle regions strongly confirmed that **on the surface of the 10Ni/TiO₂-Ref1 catalyst, large-sized Ni particles was not coated by the TiO₂-Ref1 support** (Figure R37). The distinct-different structure (encapsulated by TiO₂ and exposed) of Ni particles on 10Ni/TiO₂-OH and 10Ni/TiO₂-Ref1 indicated that SMSI in the two system was different and the Ni particles were more easily encapsulated by the TiO₂-OH support. **The Figure R35 has been shown in supplementary information as Figure S13. The Figure R36 has been shown in supplementary information as Figure S14. The Figure R37 has been shown in supplementary information as Figure S16. And the corresponding description has been shown in the revised manuscript on Page 7, lines 11–19 (highlight in yellow).**

Figure R35. (a) HRTEM and (b, c) HAADF-STEM images of the 10Ni/TiO₂-Ref1 catalyst. (d) The EDS-mapping results of the 10Ni/TiO₂-Ref1 catalyst.

Figure R36. (a) HETEM images and (b) statistical analysis of particle size for Ni particles in Ni/TiO₂-Ref1 after the RWGS reaction.

Figure R37. (a, b) The EELS-mapping results of the used 10Ni/TiO₂-Ref1 catalysts.

To investigate the disparity in the encapsulation phenomena, H₂-TPR experiments (Figure R38a) and the corresponding actual hydrogen consumption (Figure R38b) were shown. The actual hydrogen consumption of the Ni/TiO₂-OH catalyst was more than that the theoretical hydrogen consumption calculated by the reduction of NiO, suggesting that more surface oxygen of TiO₂-OH support was involved in the reduction. However, the actual hydrogen consumption of Ni/TiO₂-Ref1 was close to the theoretical hydrogen consumption, confirming that the weak MSI between Ni species and TiO₂-Ref1 (Figure R38b), which resulted in the inability of the TiO₂-Ref1 to cover even large Ni particles through strong metal-support interactions (SMSI). Therefore, the encapsulation of Ni large particles cannot be generalized, as it is closely related to the properties of the oxide supports. In addition, Ni clusters was not covered by TiO₂-OH support, since the results of *in situ* infrared spectra for 10Ni/TiO₂-OH after the RWGS reaction demonstrated CO molecules could adsorb on Ni⁰ located at 2093 cm⁻¹ and Ni²⁺ located at 2152 cm⁻¹ (Figure R39). **The Figure R38a has been shown in revised manuscript as Figure 5a. The Figure R38b has been shown in supplementary information as Figure S28. The corresponding description has been added on Page 13, lines 19–27 in the revised manuscript (highlight in yellow).**

The Figure R39 has been shown in the revised manuscript as Figure 5c with the corresponding description on Page 15, lines 7–15 (highlight in yellow).

Therefore, the disparity in the encapsulation phenomenon of these two catalysts arose from the reducibilities of TiO₂ supports and the metal-support interaction, rather than the surface energy difference. We have revised the manuscript based on experimental facts.

Figure R38. (a) H₂-TPR profiles of the NiO, 10Ni/TiO₂-Ref1 and 10Ni/TiO₂-OH samples. (b) H₂ consumption calculation from H₂-TPR of Ni/TiO₂-OH and Ni/TiO₂-Ref1 in (a).

Figure R39. *In situ* infrared spectra recorded after exposing the 10Ni/TiO₂-OH catalysts to CO with different partial pressure at 130 K after RWGS reaction.

Comment 4: “Therefore, the SMSI effect in supported Ni catalysts prevents the activation of reactant molecules at 42 the active metal sites and restricts its application in the RWGS reaction. 11,14,15” -- actually no, not necessarily, as SMSI can be used to tune selectivity. See relevant literature: 10.1126/science.adf6984, <https://doi.org/10.1038/s41467-019-12858-3>

Responses: Thanks for the reviewer’s valuable comment. We agree that SMSI does not always lead to the unsatisfactory catalytic performance. Actually, SMSI is a complex topic and it will play different roles in different catalyst systems (F. X. Xiao et al. *ACS Catal.* **2023**, 13, 10500–10510; Weckhuysen, B. M. et al. *Science* **2023**, 380, 644–651; J. Wang, et al. *ACS Catal.* 2021, 11, 6081–6090; Christopher, M. et al. *Nat. Chem.* **2017**, 9, 120–127; Bao, X. H. et al. *J. Am. Chem. Soc.* **2022**, 144, 11, 4874–4882). Based on valuable comments provided by all reviewers, we have revisited the relationship between the size effect of Ni species, SMSI and the high catalytic performance over 10Ni/TiO₂-OH. We confirmed that the high CO₂ conversion, high CO selectivity and excellent stability of 10Ni/TiO₂-OH were attributed to the high-density and stable Ni clusters, rather than the SMSI. Actually, large-sized Ni particles (tens to hundreds of nanometers) on Ni/TiO₂-OH were hardly to catalyze the CO₂ hydrogenation and promote the CH₄ selectivity. Therefore, in this work, the coverage of TiO_x on large-sized Ni particles was not related to the enhanced catalytic activity. As a result, **we have carefully revised the title, abstract and introduction of this work and we believe that the current version will deepen the previous understanding of CO₂ hydrogenation over Ni-based catalysts, cluster catalysts construction and the metal-support interaction.** Thanks for the reviewer’s comment again.

Comment 5: “uncovered by TiO₂ under RWGS reaction conditions,” -- based on what proof?

Responses: We thank for the reviewer’s comment. It was well known that the adsorption of CO was hindered when metals were encapsulated by oxide supports,

which was also one of the effective means to determine the occurrence of SMSI. To confirm whether the Ni clusters were encapsulated by TiO₂, *in situ* infrared spectra recorded after exposing the 10Ni/TiO₂-OH catalysts to CO with different partial pressure at 130 K after the RWGS reaction was conducted (Figure R39). Three bands at ~2170, 2152, and 2093 cm⁻¹ were observed. The former was available on pristine TiO₂ and attributed to CO adsorption on Ti⁴⁺ sites. The bands of 2152 cm⁻¹ and 2093 cm⁻¹ were assigned to CO linearly adsorbed on Ni²⁺ and Ni⁰ sites, respectively. The coexistence of Ni²⁺ and Ni⁰ implied the structure of clusters. Although we could not obtain direct evidence under the actual RWGS reaction conditions, the above results could indicate that Ni clusters were uncovered after the *in situ* RWGS reaction. **The Figure R39 has been shown in the revised manuscript as Figure 5c with the corresponding description on Page 15, lines 7–15 (highlight in yellow).**

Comment 6: “In our discovery, this was also proven by the fact that the larger Ni particles were encapsulated, while the Ni clusters were not. “ -- but it was already known, that NPs are more prone; why do the authors claim this is something new, that it is a “discovery”?”

Responses: Thanks for reviewer’s valuable comments. We have deleted the statement of “In our discovery”.

Comment 7: The reaction rate was 10 times higher, which suggests differing mechanisms; further suggesting that the actual underlying reason for the observed differences must better be corroborated (e.g., size vs MSI effect vs defect sites on treated TiO₂).

Responses: Thanks for reviewer’s valuable comments. In order to respond to the reviewer’s comment, we have done additional experiments and analyzed the results as below:

1. The influence of particle size:

The particle size of Ni species in Ni/TiO₂-OH after the H₂ pretreatment and the following RWGS reaction was ~1 nm (Figure R40a, d), while Ni species in Ni/TiO₂-Ref1 were sintered into ~27.6 nm (Figure R40b, e). In order to increase the gradient of the particle size distribution, we reduced the temperatures of air calcination and H₂ pretreatment of Ni/TiO₂-Ref1 from 600 °C to 500 °C (the obtained sample was denoted as “Ni/TiO₂-Ref-500”). TEM images results showed that the average size of Ni particles on Ni/TiO₂-Ref-500 was ~12.4 nm (Figure R40c, f). As illustrated in Figure R41a and b, the CO₂ conversion and reaction rate ranked in the order of Ni/TiO₂-OH > Ni/TiO₂-Ref1-500 > Ni/TiO₂-Ref1, which was opposite to the ordering of the Ni size. In order to compare the CO selectivity of these three samples with different Ni sizes in a more rational way, we varied the gas hourly space velocity (GHSV) to regulate the comparability of CO₂ conversion (Figure R41c). The CO selectivity of Ni/TiO₂-Ref1-500 with Ni particle size of 12.4 nm was only 21.3%, much lower than that of Ni/TiO₂-OH with Ni cluster size of 1.0 nm, which was consistent with the notion that larger particle sizes tend to be more active for methanation (Christopher, P. et al. *J. Am. Chem. Soc.* **2015**, 137, 3076–3084; Guo, L. M. et al. *ACS Catal.* **2023**, 13, 10364–10374). However, as the Ni size increased to 27.6 nm, the CO₂ conversion could not be tuned to a comparable level even when the GHSV was lowered to 20,000 mL·g_{cat}⁻¹·h⁻¹, suggesting that the activity of Ni/TiO₂-Ref1 was very poor and the CO selectivity was meaningless high.

Based on above results, the relationship between the Ni size on TiO₂, SMSI and the catalytic performance (CO₂ conversion and product selectivity) in atmospheric CO₂ hydrogenation could be clearly revealed (Figure R42). **On the one hand, compared to highly dispersed Ni clusters, Ni particles induced the formation of CH₄. On the other hand, when the Ni particle size was larger than ~27 nm, it was meaningless to discuss the product selectivity due to the very low activity.**

The Figure R40a has been shown in revised manuscript as Figure 2b. The Figure R40d has been shown in supplementary information as Figure S8c. The Figure R40b, e has been shown in supplementary information as Figure S22. The Figure

R40c, f has been shown in supplementary information as Figure S14. Figure R41a, b has been shown in supplementary information as Figure S23a, b. Figure R41c has been shown in revised manuscript as Figure 4b. Figure R42 has been shown in revised manuscript as Figure 4c. The corresponding description has been added on Page 10, lines 9–28 and Page 11, line 1 in the revised manuscript (highlight in yellow).

Figure R40. (a) HAADF-STEM images of Ni/TiO₂-OH after H₂ pretreatment and RWGS at 600 °C and (b, c) TEM images of Ni/TiO₂-Ref1 after H₂ pretreatment and RWGS at 500 °C (b) and 600 °C (c). (d–f) The statistical analysis of particle size for Ni particles in (a–c), respectively.

Figure R41. (a) The CO₂ conversion and CO selectivity for 10Ni/TiO₂-OH, 10Ni/TiO₂-Ref1 (air calcination and H₂ pretreatment at 600 °C) and 10Ni/TiO₂-Ref1-500 (air

calcination and H₂ pretreatment at 500 °C) under the GHSV of 400,000 mL·g_{cat}⁻¹·h⁻¹. (b) Comparison of CO yield rates over these three samples in (a) at 500 °C. (c) The CO₂ conversion and CO selectivity for these three samples in (a) at 500 °C under different GHSVs.

Figure R42. The schema of the influence of Ni species size on the catalytic performance over Ni/TiO₂ catalysts.

2. The influence of metal-support interaction (MSI):

It has been established that catalyst structures and catalytic performances are largely determined by the metal-support interaction (MSI), which was closely related to the properties of the support. Through the characterization of H₂-TPR, *quasi in situ* XPS, *in situ* infrared spectra and the *in-situ* Ni L-edge NAP-NEXAFS profile in Figure 5 in revised manuscript, the MSI over Ni/TiO₂-OH was much stronger than that in Ni/TiO₂-Ref1, which was the fundamental reason why Ni clusters anchored on TiO₂-OH could still be maintained even under the RWGS reaction with high operating temperatures and reducing atmospheres. It should be noted that the hydroxylation of the commercial TiO₂ was the prerequisite to disperse Ni species as isolated atoms, which guaranteed the formation of dense and stable Ni clusters during the RWGS reaction.

3. The influence of surface defect sites on treated TiO₂:

To investigate the effect of defect sites of the treated TiO₂ in the RWGS reaction, we employed various characterizations to confirm the low content of surface oxygen vacancies. Firstly, as displayed by XPS Ti 2*p* spectra of as-prepared and used Ni/TiO₂-OH and Ni/TiO₂-Ref1, only the presence of Ti⁴⁺ was observed at 458.5 eV (Figure R33). Moreover, only Ti⁴⁺ was detected in the Ti L₃ edge near ambient pressure (NAP) – near edge X-ray absorption fine structure (NEXAFS) under H₂ and CO₂+ H₂ conditions (Figure R34), consistent with the XPS results. Based on above results, we speculated that even though the MSI between Ni and TiO₂-OH facilitated the partial reduction of TiO₂, the limited concentration of surface Ti³⁺ sites were below the detection line of XPS and NEXAFS techniques. According to previous report, the absence of surface oxygen vacancies might be caused by the easy mobility of oxygen vacancies from surface to bulk on anatase TiO₂ (Scheiber, P. et al. *Phys. Rev. Lett.* **2012**, 109, 136103). Therefore, we thought that the enhanced catalytic activity was not related to the surface oxygen vacancies on the treated TiO₂. **The Figure R33, 34 has been shown in supplementary information as Figure S32 with the corresponding discussion on Page 15, lines 27, 28 and Page 16, lines 1, 2 in the revised manuscript.**

Combining the three aspects mentioned above, we could conclude that **the hydroxylation of TiO₂, the high-density and stable Ni clusters, as well as the strong interaction between Ni species and TiO₂-OH** were the key factors that attributed to the unmatched catalytic performance of Ni/TiO₂-OH. Thanks for reviewer's comments again.

***Comment 8:** There is no blank with TiO₂-OH; how can the authors say their work performs better when it in theory could simply be the TiO₂ which becomes more active? The TiO₂ (tube) precursor formation could severely affect the electronic properties of the TiO₂, and its effect should be evaluated.*

Responses: Reviewer's valuable comments are appreciated by us. Based on reviewer's comments, we added the activity evaluation of the TiO₂-OH. As shown in Figure R43, TiO₂-OH almost had no catalytic activity even at 600 °C, indicating that the synergy of

Ni species and TiO₂-OH promoted the RWGS reaction. Thanks for the reviewer's comment.

Figure R43. The CO₂ conversion and CO selectivity of TiO₂-OH.

Comment 9: “we developed a strategy to fabricate SMSI-resistant Ni clusters by reducing intrinsic hydroxyl-anchored isolated Ni atoms on the TiO₂ surface.” -- is it that, or is it simply the particle size preventing encapsulation which was known already?

Responses: Thanks for the reviewer's valuable comments. We carefully reviewed the whole manuscript based on the reviewer's comments. Actually, the phenomenon that Ni clusters on TiO₂-OH were not encapsulated by the TiO₂ support was consistent with previous research results, which was attributed to the size effect of the metal particles.

However, in our work, large-sized Ni particles were encapsulated by TiO₂-OH but not by TiO₂-Ref1, indicating that the encapsulation of large metal particles was not a universal phenomenon. H₂-TPR experiments (Figure R44a) and the corresponding actual hydrogen consumption (Figure R44b) showed that the actual hydrogen consumption of Ni/TiO₂-OH catalyst was more than the theoretical hydrogen consumption of NiO, suggesting that more surface oxygen atoms of TiO₂-OH support were reduced under H₂ flow. While the actual hydrogen consumption of Ni/TiO₂-Ref1 was close to the theoretical hydrogen consumption, confirming that the weak MSI between Ni species and TiO₂-Ref1 (Figure R44b). Therefore, the occurrence of the

encapsulation phenomenon is not only related to the size of Ni particle but also to the reducibility of the TiO₂ support.

Figure R44. (a) H₂-TPR profiles of the NiO, 10Ni/TiO₂-Ref1 and 10Ni/TiO₂-OH samples. (b) H₂ consumption calculation from H₂-TPR of Ni/TiO₂-OH and Ni/TiO₂-Ref1 in (a).

References

1. Scheiber, P. *et al.* Surface mobility of oxygen vacancies at the TiO₂ anatase (101) surface. *Phys. Rev. Lett.* **109**, 1–5 (2012).
2. Wang, H. *et al.* Construction of Pt^{δ+}-O(H)-Ti³⁺ Species for Efficient Catalytic Production of Hydrogen. *ACS Catal.* **13**, 10500-10510 (2023).
3. Monai, M. *et al.* Restructuring of titanium oxide overlayers over nickel nanoparticles during catalysis. *Science* **380**, 644–651 (2023).
4. Matsubu, J. C. *et al.* Adsorbate-mediated strong metal–support interactions in oxide-supported Rh catalysts. *Nat. Chem.* **9**, 120–127 (2017).
5. Xin, H. *et al.* Overturning CO₂ Hydrogenation Selectivity with High Activity via Reaction-Induced Strong Metal–Support Interactions. *J. Am. Chem. Soc.* **144**, 11, 4874–4882 (2022).
6. Matsubu, J. C. *et al.* Isolated Metal Active Site Concentration and Stability Control Catalytic CO₂ Reduction Selectivity. *J. Am. Chem. Soc.* **137**, 3076–3084 (2015).

7. Yang, B. et al. Size-Dependent Active Site and Its Catalytic Mechanism for CO₂ Hydrogenation Reactivity and Selectivity over Re/TiO₂. *ACS Catal.* **13**, 10364-10374 (2023).

REVIEWER COMMENTS

Reviewer #1 (Remarks to the Author):

I have now read the responses to the three referee report and based on the revised article and the revisions/additions made to the initial criticism and suggestions I am pleased to recommend the work for publication in Nature Communications. My own comments have also well addressed by the authors.

Reviewer #2 (Remarks to the Author):

Authors answered my concerns.

Reviewer #3 (Remarks to the Author):

The authors have reported the change of product selectivity of CO₂ hydrogenation from CH₄ to CO using TiO₂ supported Ni catalyst, breaking the activity-selectivity trade-off of the reaction. It is an amazing result to keep Ni at nanocluster states at a reduction temperature above 500 °C and a loading ratio of 10%. Given that the authors have seriously revised the current manuscript and provided more supporting data, and that have addressed the comments raised by the reviewers, I think it is suitable to accept this paper for publication on Nature Communications after adding some further discussions on the reaction mechanisms.

The authors have provided some DFT calculation results to discuss the reaction mechanism, while I notice that they have also performed some in situ FTIR measurements in Figure S35. I suggest the authors to move these data to the main text as experimental supports of their arguments and further discuss the FTIR results to support their conclusions.

Reviewer #4 (Remarks to the Author):

I have reread the authors revised manuscript. The authors did a good job addressing the comments of the reviewers, and I do believe that the article, after major revisions may be suitable for publication.

“can be dispersed as isolated Ni atoms” >>> no proof for that was provided. Rephrase.

“by green hydrogen to form CO” >>> strange phrasing in sentence. Not only green hydrogen can convert CO₂.

“by the severe methanation” >>> remove “the”, also in following sentence.

Although the methanation can be effectively inhibited through reported methods including reducing metal loading,^{10–12} forming metal carbide/phosphide^{2,13} and inducing the encapsulation layer over metal sites,^{14–16} a large number of active sites are inevitably lost during above processes, >>> as well as decreasing particle size, and varying the support material (Vogt, C. et al. Nat. Comm. 2019, 10, 5330, Vogt, C. et al. Nat. Catal. 2018, 1, 127-134). For those, no active sites are lost. Actually, they are gained in making smaller nanoparticles.

On the other hand, active sites possess weak CO adsorption capacity >>> rephrase to: On the other hand, the active sites must possess relatively

flexible surface oxygen atoms >>> the atoms are flexible ...? Rephrase.

“Unfortunately, due to the moderate metal-support interaction (MSI)” >>> again “due to the moderate MSI”; TiO₂ is typically believed to have strong metal support interaction, especially with clusters. You must thus further specify what exactly you mean.

is to build an enhanced >>> Rephrase to “is to induce enhanced...”

catalyst exceeds almost all reported catalysts, >>> catalyst towards CO₂ reduction to CO

I miss an overview of “all catalysts reported” in literature that the authors compare their data to, both as figure, and as table with references in the supporting information.

10Ni/TiO₂-OH-UC >>> when you first introduce the term, explain what UC refers to.

“Accompanied by the dehydration of H₂Ti₃O₇ to anatase TiO₂ via the air calcination” >>> unclear what you mean here with accompanied by the dehydration. Do you mean an additional sample was made? Then state that. How was that sample prepared then if not via deposition precipitation?

Lots of grammar and spelling issues, paper needs to be redacted professionally:
which was consisted with the XRD result (Supplementary Fig. 12). In contrast, >>> consistent. In contrast.

distinct-different >>> distinctly different

Meanwhile, 10Ni/TiO₂-OH showed excellent CO selectivity, maintaining above 90% at all reaction temperatures, which reversed the conventional understanding that Ni catalysts are more suitable for methanation >>> I would phrase it a bit less strongly, as indeed, as you cite, it was already known that CO selectivity can dramatically go up with smaller particle size. I would phrase this as: “, in clear contrast to the generally observed near 100% selectivity towards methane which is generally observed on nickel-based catalysts.”

and name it 10Ni/TiO₂-Ref1-500 >>> “referred to as 10Ni/TiO₂-Ref1-500”

“Based on above results, the relationship between the Ni size on TiO₂ and the product selectivity in

atmospheric CO₂ hydrogenation could be clearly revealed (Fig. 2c).” >>> Figure 4C and the relationship in general. The authors have (also, with only 3 data points..., and also with different OH dominance on the reaction surface) not proven causation of the high selectivity towards CO on the cluster-based catalyst. Correlation, at best, but who is to say that the very high activity towards CO is not based predominantly on the support characteristics? It is misleading to show a figure as in 4C, then. I.e., is the conversion happening solely on the Ni? I believe not; Ti/OH can actively participate in the reaction and the relative amount of interface sites is more likely to be the point here.

10Ni/TiO₂-OH remained stable with the CO₂ conversion more than ~55% and showed almost complete CO selectivity. In contrast, under relative milder reaction conditions, 10Ni/TiO₂-Ref1 lost almost all of its activity in only 2.5 hours (Supplementary Fig. 26a). >>> The authors should discuss the previous comment, also in line with this observation here. What is the particle size after spent analysis in this long term stability test? Is the particle size increase accounting for the loss in

“was close to the theoretical hydrogen consumption, confirming that the weak MSI between Ni species and TiO₂-Ref1 (Figure R38b), which resulted in the inability of the TiO₂-Ref1 to cover even large Ni particles through strong metal-support interactions (SMSI).” >>> Or hydrogen spillover is easier onto that TiO_x...

Response to reviewers: “After the RWGS reaction at 600 °C, the Ni particles further aggregated into larger size of about 27.6 nm (Figure R36). The EELS-mapping results of two different Ni particle regions strongly confirmed that on the surface of the 10Ni/TiO₂-Ref1 catalyst, largesized Ni particles was not coated by the TiO₂-Ref1 support (Figure R37).” >>> Are these in situ? After exposure to oxygen it's not said that they will remain. Ex-situ images lacking proof of encapsulation don't prove anything on what happens in situ.

Figure R32: Without fits, how can the Ni-Ti be indicated? Or NiO or Ni-Ni for that matter, although in the reference materials I can live with that.

“uncovered by TiO₂ under RWGS reaction conditions,” >>> change to not covered, uncovered here is a bit ambiguous.

“CO molecules could adsorb on Ni⁰ located at 2093 cm⁻¹ and Ni²⁺ located at 2152 cm⁻¹ (Figure R39). The Figure R38a has been shown in revised manuscript as Figure 5a.” >>> The peak assignment in Figure 5C should be further discussed. 2090 cm⁻¹ is typically already sub-carbonyl species on Ni, Ni tetra carbonyl can even form. 2060 is more in line with Ni⁰ on NPs.

Fig R39: 10Ni/TiO₂-OH catalysts to CO with different partial pressure at 130 K after RWGS reaction. >>> how meaningful are these tests/the results? Clearly, the active catalyst is expected to have a relatively low CO adsorption strength. CO adsorption measurements where the pressure is different to that of the reaction conditions thus are not really meaningful; please discuss the results in light of that statement and also the previous comment. Also, the way the figure is ordered is strange, why not from low to high pressure, instead of from low to high to lower? And why not list

CO in the top figure as in the bottom 2?

“The former signal related to the CO adsorption on Ti⁴⁺ sites was also found on pure TiO₂ 52 (Supplementary Fig. 30b).” >>> what about IR of the TiO₂-OH sample? I find the 2152 to be dubious and it must be ruled out that it is not related to the TiO₂.

10Ni/TiO₂-OH remained stable with the CO₂ conversion more than ~55% and showed almost complete CO selectivity. In contrast, under relative milder reaction conditions, 10Ni/TiO₂-Ref1 lost almost all of its activity in only 2.5 hours (Supplementary Fig. 26a). >>> why are the reference and main cat not compared under the same long-term stability conditions?

“The strong interaction between intrinsic OH groups was proven”

References missing in main text: F. X. Xiao et al. ACS Catal. 2023, 13, 10500–10510; Vogt, C. et al. Nat. Comm. 2019, 10, 5330; J. Wang, et al. ACS Catal. 2021, 11, 6081–6090; Bao, X. H. et al. J. Am. Chem. Soc. 2022, 144, 11, 4874–4882.

Responses to the Reviewers' Comments and the Corresponding Revisions

Reviewer #1: *I have now read the responses to the three referee report and based on the revised article and the revisions/additions made to the initial criticism and suggestions I am pleased to recommend the work for publication in Nature Communications. My own comments have also well addressed by the authors.*

Response: Thanks for reviewer's comments. According to reviewer's comments, we have carefully revised and supplemented the manuscript, which undoubtedly greatly improved the quality of our research work. Thanks for the reviewer again.

Reviewer #2: *Authors answered my concerns.*

Response: The reviewer's professional and detailed comments and suggestions are of great help to the improvement of our research work. According to the reviewer's comments, the quality of our manuscript has been greatly improved compared to the original version. Thanks for the reviewer again.

Reviewer #3: *The authors have reported the change of product selectivity of CO₂ hydrogenation from CH₄ to CO using TiO₂ supported Ni catalyst, breaking the activity-selectivity trade-off of the reaction. It is an amazing result to keep Ni at nanocluster states at a reduction temperature above 500 °C and a loading ratio of 10%. Given that the authors have seriously revised the current manuscript and provided more supporting data, and that have addressed the comments raised by the reviewers, I think it is suitable to accept this paper for publication on Nature Communications after adding some further discussions on the reaction mechanisms.*

Response: We sincerely thank you for your thorough evaluation and examination of our work. Based on reviewer's comments, we replenish further discussions on the reaction mechanisms.

Comment 1: *The authors have provided some DFT calculation results to discuss the reaction mechanism, while I notice that they have also performed some in situ FTIR measurements in Figure S35. I suggest the authors to move these data to the main text*

as experimental supports of their arguments and further discuss the FTIR results to support their conclusions.

Response: Thanks for reviewer's valuable comments. As shown in Figure R1, we have moved the data of *in situ* diffused reflectance infrared Fourier transform spectroscopy (DRIFTS) spectra to the revised manuscript as Figure 6b and 6c. The further discussion has been added on pages 16–18 (highlight in yellow). Thanks for reviewer's comments again.

Figure R1. (a) CO₂ adsorption on Ni₈/TiO₂ system and H₂ adsorption on CO₂-Ni₈/TiO₂ system. (b, c) *In situ* diffused reflectance infrared Fourier transform

spectroscopy (DRIFTS) spectra of 10Ni/TiO₂-OH catalyst during CO₂ treatment and reaction conditions (CO₂+H₂) at 300 °C, respectively. (d) Potential energy diagram for the RWGS reaction on Ni₈/TiO₂(101) via carboxyl and formate pathways. The illustration showed the structures of some intermediate and transition states. The complete reaction pathways are shown in Supplementary Fig. 39 (carboxyl) and 40 (formate). “TS” represents transition state.

Reviewer #4: *I have reread the authors revised manuscript. The authors did a good job addressing the comments of the reviewers, and I do believe that the article, after major revisions may be suitable for publication.*

Response: Reviewer’s professional and detailed comments are very important for improving our research work. Based on reviewer’s comments, we further revised the manuscript point by point.

Comment 1: *“can be dispersed as isolated Ni atoms” >>> no proof for that was provided. Rephrase.*

Response: We are grateful for the reviewer’s comment. We have rephrased this as “**can be highly dispersed** on hydroxylated TiO₂” (line 4, page 2, highlight in yellow).

Comment 2: *“by green hydrogen to form CO” >>> strange phrasing in sentence. Not only green hydrogen can convert CO₂.*

Response: Reviewer’s comment is highly agreed by us. We have rephrased this sentence as “As an available but inert source of carbon, **greenhouse gas CO₂ can be converted by hydrogen to form CO**, which can be served as a crucial pathway to synthesize liquid fuel by coupling syngas processes.” (lines 2 and 3, page 3, highlight in yellow). Thanks for the reviewer’s comment again.

Comment 3: *“by the severe methanation” >>> remove “the”, also in following sentence.*

Response: Thank you for your careful correction. We have removed “the” in the related sentences.

Comment 4: Although the methanation can be effectively inhibited through reported methods including reducing metal loading,10–12 forming metal carbide/phosphide^{2,13} and inducing the encapsulation layer over metal sites,14–16 a large number of active sites are inevitably lost during above processes, >>> as well as decreasing particle size, and varying the support material (Vogt, C. et al. Nat. Comm. 2019, 10, 5330, Vogt, C. et al. Nat. Catal. 2018, 1, 127-134). For those, no active sites are lost. Actually, they are gained in making smaller nanoparticles.

Response: Thanks for reviewer’s valuable comments. According to reviewer’s comment, we have added relevant strategies and cited the references mentioned. The corresponding description has been modified as “Currently, methanation can be effectively inhibited through reported methods including reducing metal loading (*Li, S. et al. Angew. Chem. Int. Ed.* **2017**, 129, 10901–10905; *Dong, C. et al. J. Am. Chem. Soc.* **2023**, 145, 17056–17065; *Yang, B. et al. ACS Catal.* **2023**, 13, 10364–10374.), forming metal carbide/phosphide (*Galhardo, T. S. et al. J. Am. Chem. Soc.* **2021**, 143, 4268–4280; *Wei, X. et al. J. Am. Chem. Soc.* **2023**, 145, 14298–14306), inducing the encapsulation layer over metal sites (*Li, J. et al. ACS Catal.* **2019**, 9, 6342–6348; *Matsubu, J. C. et al. Nat. Chem.* **2017**, 9, 120–127; *Xin, H. et al. J. Am. Chem. Soc.* **2022**, 144, 4874–4882), decreasing particle size (*Vogt, C. et al. Nat. Catal.* **2018**, 1, 127–134), and varying the support material (*Vogt, C. et al. Nat. Commun.* **2019**, 10, 1–10). However, the lack of active sites in above processes leads to a significant activity trade-off for high CO selectivity.” (lines 6–10, page 3, highlight in yellow).

Comment 5: On the other hand, active sites possess weak CO adsorption capacity >>> rephrase to: On the other hand, the active sites must possess relatively

Response: We appreciate the reviewer for this comment. We have revised this sentence on line 14, page 3 (highlight in yellow).

Comment 6: flexible surface oxygen atoms >>> the atoms are flexible ...? Rephrase.

Response: Thanks to the reviewer for this valuable suggestion. We have rephrased this to “As one typical reducible support, TiO₂ with **active surface oxygen** has been widely used in supporting active metals to catalyze the RWGS reaction.” on line 20, page 3.

Comment 7: “Unfortunately, due to the moderate metal-support interaction (MSI)” >>> again “due to the moderate MSI”; TiO₂ is typically believed to have strong metal support interaction, especially with clusters. You must thus further specify what exactly you mean.

Response: We appreciate this comment raised by the reviewer. In fact, while TiO₂ exhibits typical strong metal-support interactions, its ability to disperse and anchor metal species is limited by the loading amount of the metal (Wang, C. *et al. J. Am. Chem. Soc.* 2024, 146, 8, 5523–5531; Li, X. *et al. Angew. Chem. Int. Ed.* 2020, 59, 19983–19989). A feasible pathway to overcome above challenges is to construct anchoring sites on the TiO₂ surface to suppress the sintering of high-density clusters during the reaction. Therefore, we have revised the wording to “Unfortunately, **due to its limited ability to disperse metal species**” and “A feasible pathway to overcome above challenges is to **construct anchoring sites on the TiO₂ surface** to suppress the sintering of high-density clusters during the reaction.” on lines 21, 22 and 24, 25, page 3.

Comment 8: is to build an enhanced >>> Rephrase to “is to induce enhanced...”

Response: Thanks for the reviewer’s valuable comment. We have revised this expression on line 24, page 3 (highlight in yellow).

Comment 9: catalyst exceeds almost all reported catalysts, >>> catalyst towards CO₂ reduction to CO

Response: Thank the reviewer for this suggestion. We have revised the sentence on line 11, page 4 (highlight in yellow).

Comment 10: I miss an overview of “all catalysts reported” in literature that the authors compare their data to, both as figure, and as table with references in the supporting information.

Response: Thanks a lot for the important suggestion. Table R1 and Figure R2 summarized **the comparison of reaction rate and CO selectivity between the catalyst in this work and the reported catalysts**, which was shown as Supplementary Table 3 and Figure 4c in the revised SI and manuscript, respectively.

Table R1. Comparison of CO₂ conversion rate and CO selectivity for the as-prepared and literature reported catalysts.

Catalyst	H ₂ :C O ₂	Temperatu re (°C)	Pressur e (MPa)	Rate ($\mu\text{mol}_{\text{CO}}/\text{g}_{\text{cat}}/\text{s}$)	CO ₂ Conv. rate ($\mu\text{mol}/\text{g}_{\text{cat}}/\text{s}$)	CO selectivity (%)	Ref
10Ni/TiO ₂ - OH	3:1	500	0.1	873.0 ^b	—	98.0	this work
10Ni/TiO ₂ - OH	3:1	400	0.1	—	214.4 ^b	91.9	this work
(1) Ni/ MAO1000	4:1	400	0.1	—	53.3 ^a	3	18
(2) 10Co/ r-TiO ₂	4:1	400	3	—	19.16 ^a	2	19
(3) Ni ₃ Fe ₂ /ZrO ₂	2:1	400	0.1	—	13.86 ^a	11.5	20
(4) Ni ₃ Fe ₃ /ZrO ₂	2:1	400	0.1	—	13.57 ^a	12.9	20
(5) Ni ₃ Fe ₁ /ZrO ₂	2:1	400	0.1	—	13.22 ^a	14.0	20
(6)	2:1	400	0.1	—	12.6 ^a	15.3	20

Ni ₃ /ZrO ₂							
(7) Ni-P-12	4:1	400	0.1	—	1.83 ^a	19.7	21
(8) 2Ni- 4nm /CeO ₂	3:1	400	0.1	—	74.4 ^b	25.0	22
(9) Ni-P-8.7	4:1	400	0.1	—	74.4 ^a	70.0	21
(10) Rh@S- 1	3:1	400	1	—	3.87 ^a	82.7	23
(11) 10Co/ a-TiO ₂	4:1	400	3	—	3.33 ^a	90	19
(12) Ni ₃ Fe ₉ /ZrO ₂	2:1	400	0.1	—	6.4 ^b	95.8	20
(13) Ni-P- 4.2	4:1	400	0.1	—	1.49 ^a	96.6	21
(14) NiIn 0.5 /SBA-15	4:1	400	0.1	—	13.2 ^a	99.0	24
(15) 2Ni- 4nm /Mo ₂ N	3:1	400	0.1	—	134.0 ^b	99.9	22
(16) Cu /β-Mo ₂ C	2:1	500	0.1	379.0 ^a	—	99.0	25
(17) Cu /CeO ₂ -hs	3:1	500	0.1	334.7 ^a	—	100	26
(18) Cu- Zn-Al	2:1	500	0.1	261.0 ^a	—	100	26
(19) Ni in Cu	3:1	500	0.1	39.5 ^a	—	100	27

^aOut of the kinetic interval

^bIn the kinetic interval

Figure R2. CO₂ conversion rate versus CO selectivity reported non-Cu based catalyst for CH₄ production (blue triangles, Sel_{CO}<50 %) and CO production (red squares, Sel_{CO}>50 %), and this work (red circle) at 400 °C. Inset: comparison of CO yield rates over 10Ni/TiO₂-OH catalysts and Cu-based catalysts at 500 °C.

Comment 11: 10Ni/TiO₂-OH-UC >>> when you first introduce the term, explain what UC refers to.

Response: Thanks for this valuable suggestion. The 10Ni/TiO₂-OH-UC sample was prepared by deposition-precipitation method without further air calcination. We have added a further explain of “**without air calcination**” on line 24, page 4 (highlight in yellow).

Comment 12: “Accompanied by the dehydration of H₂Ti₃O₇ to anatase TiO₂ via the air calcination” >>> unclear what you mean here with accompanied by the dehydration. Do you mean an additional sample was made? Then state that. How was that sample prepared then if not via deposition precipitation?

Response: Thank the reviewer for this question. No additional samples were made. The

deposition-precipitation method was used to load 10 wt. % Ni on $\text{H}_2\text{Ti}_3\text{O}_7$ support with intrinsic $-\text{OH}$ groups, and the obtained sample was denoted by $10\text{Ni}/\text{TiO}_2\text{-OH-UC}$. **The catalyst was subsequently calcinated in air at $600\text{ }^\circ\text{C}$, during which $\text{H}_2\text{Ti}_3\text{O}_7$ with intrinsic $-\text{OH}$ groups underwent dehydration, leading to a phase change to anatase TiO_2 .** Nevertheless, **partial $-\text{OH}$ groups still retained on the catalyst surface** (Figure R1). Meanwhile, Ni was still dispersedly anchored on the $\text{TiO}_2\text{-OH}$ support (denoted by $10\text{Ni}/\text{TiO}_2\text{-OH}$). Figure R3 was shown in the supplementary information as Figure S4. **We have added further statement on line 26, page 4 and lines 6, page 5.**

Figure R3. The ATR-FTIR spectra of $10\text{Ni}/\text{TiO}_2\text{-OH}$ and $10\text{Ni}/\text{TiO}_2\text{-Ref1}$ samples.

Comment 13: Lots of grammar and spelling issues, paper needs to be redacted professionally: which was consisted with the XRD result (Supplementary Fig. 12). In contract, >>> consistent. In contrast. distinct-different >>> distinctly different

Response: Thank the reviewer for pointing out our errors. We carefully checked the grammar and spelling issues in the article and made corrections.

Comment 14: Meanwhile, $10\text{Ni}/\text{TiO}_2\text{-OH}$ showed excellent CO selectivity, maintaining above 90% at all reaction temperatures, which reversed the conventional understanding that Ni catalysts are more suitable for methanation >>> I would phrase

it a bit less strongly, as indeed, as you cite, it was already known that CO selectivity can dramatically go up with smaller particle size. I would phrase this as: “, in clear contrast to the generally observed near 100% selectivity towards methane which is generally observed on nickel-based catalysts.”

Response: Thank the reviewer so much for this professional suggestion. We have rephrased this sentence on lines 7 and 8, page 10 (highlight in yellow).

Comment 15: *and name it 10Ni/TiO₂-Ref1-500 >>> “referred to as 10Ni/TiO₂-Ref1-500”*

Response: Thanks for the reviewer’s valuable suggestion. We have revised this sentence on line 19, page 10 (highlight in yellow).

Comment 16: *“Based on above results, the relationship between the Ni size on TiO₂ and the product selectivity in atmospheric CO₂ hydrogenation could be clearly revealed (Fig. 2c).” >>> Figure 4C and the relationship in general. The authors have (also, with only 3 data points, and also with different OH dominance on the reaction surface) not proven causation of the high selectivity towards CO on the cluster-based catalyst. Correlation, at best, but who is to say that the very high activity towards CO is not based predominantly on the support characteristics? It is misleading to show a figure as in 4C, then. I.e., is the conversion happening solely on the Ni? I believe not; Ti/OH can actively participate in the reaction and the relative amount of interface sites is more likely to be the point here.*

Response: We appreciated the reviewer’s valuable comments. Based on our experimental results, it could be confirmed that the CO₂ conversion and CO selectivity of Ni-based catalysts were closely related to the Ni size. However, as the reviewer mentioned, **the hydroxylation of TiO₂, the high-density and stable Ni clusters, as well as the strong interaction between Ni species and TiO₂-OH were all key factors** that attributed to the high catalytic performance of Ni/TiO₂-OH. It was biased and misleading to only summarize the relationship between size and reaction performance. Therefore, **we deleted this schematic and the corresponding description.** The

corrected figure was presented in Figure R4, which was shown as Figure 4 in the revised manuscript.

Figure R4. Catalytic performance for CO₂ hydrogenation. (a) The CO₂ conversion and CO selectivity for 10Ni/TiO₂-OH and 10Ni/TiO₂-Ref1 catalysts under the GHSV of 400,000 mL·g_{cat}⁻¹·h⁻¹. (b) Comparison of CO selectivity at 500 °C within relative comparable CO₂ conversion rates for 10Ni/TiO₂-OH, 10Ni/TiO₂-Ref1 and 10Ni/TiO₂-Ref1 after calcination at 500 °C and H₂ pretreatment at 500 °C. (c) CO₂ conversion rate versus CO selectivity reported non-Cu based catalyst for CH₄ production (blue triangles, Sel_{CO}<50 %) and CO production (red squares, Sel_{CO}>50 %), and this work (red circle) at 400 °C. Inset: comparison of CO yield rates over 10Ni/TiO₂-OH catalysts and Cu-based catalysts at 500 °C. (d) The long-term stability of the 10Ni/TiO₂-OH catalyst tested for 300 h at 600 °C (GHSV = 800,000 mL·g_{cat}⁻¹·h⁻¹).

Comment 17: 10Ni/TiO₂-OH remained stable with the CO₂ conversion more than ~55% and showed almost complete CO selectivity. In contrast, under relative milder reaction conditions, 10Ni/TiO₂-Ref1 lost almost all of its activity in only 2.5 hours (Supplementary Fig. 26a). >>> The authors should discuss the previous comment, also in line with this observation here. What is the particle size after spent analysis in this long term stability test? Is the particle size increase accounting for the loss in

Response: Reviewer's valuable comments are highly appreciated by us. After H₂ pretreatment, Ni species of 10Ni/TiO₂-Ref1 were agglomerated into large particles of

~18.8 nm (Figure R5a). After undergoing the stability test for 2.5 hours, Ni particles further aggregated to ~33.7 nm (Figure R5b), which indicated that the loss of activity was attributed to the increase in particle size. The above results have been added as Figure S27 in the revised SI and the corresponding discussion has been added on lines 12–14, page 12.

Figure R5. TEM images and corresponding statistical analysis of particle size for Ni particles of 10Ni/TiO₂-Ref1 catalyst after (a, b) H₂ pretreatment at 600 °C and (c, d) long term stability test.

Comment 18: “was close to the theoretical hydrogen consumption, confirming that the weak MSI between Ni species and TiO₂-Ref1 (Figure R38b), which resulted in the inability of the TiO₂-Ref1 to cover even large Ni particles through strong metal-support interactions (SMSI).” >>> Or hydrogen spillover is easier onto that TiO_x...

Response: Thanks for reviewer’s valuable questions. It was well known that the hydrogen spillover from metal nanoparticles to the oxide support could promote the

reduction of surface oxygen on the support. As the literature reported, the occurrence of hydrogen spillover could be demonstrated by the shift of surface oxygen reduction to lower temperatures in H₂-TPR experiments (Kim, H. et al. ACS Catal. 2023, 13, 11857-11870; Kim, H. et al. J. Catal. 2022, 410, 93–102). The peaks located at 356 °C and 468 °C were attributed the reduction of Ni species (Figure R6a). The similarity between the actual and theoretical hydrogen consumption confirmed no additional oxygen atoms were involved in the reduction, that was, there was no hydrogen spillover in this process. However, after the complete reduction of Ni species, the surface oxygen of TiO₂ was reduced at 609 °C, which was lower than that in pristine TiO₂-Ref1 at 671 °C. The result implied that hydrogen spillover might occur in 10Ni/TiO₂-Ref1 (Figure R6b).

Figure R6. (a) H₂-TPR profiles of the TiO₂-Ref1 and 10Ni/TiO₂-Ref1 samples. (b) An enlarged region of (a).

Comment 19: Response to reviewers: “After the RWGS reaction at 600 °C, the Ni particles further aggregated into larger size of about 27.6 nm (Figure R36). The EELS-mapping results of two different Ni particle regions strongly confirmed that on the surface of the 10Ni/TiO₂-Ref1 catalyst, large sized Ni particles was not coated by the TiO₂-Ref1 support (Figure R37).” >>> Are these in situ? After exposure to oxygen it's not said that they will remain. Ex-situ images lacking proof of encapsulation don't prove anything on what happens in situ.

Response: The reviewer’s valuable comments are highly agreed by us. Unfortunately, our electron microscopy images were all obtained under *ex-situ* conditions. *In-situ* DRIFTS experiments of CO adsorption after the RWGS reaction were carried out to indicate the exposure of Ni species. At $-20\text{ }^{\circ}\text{C}$ and without exposure to air, the CO adsorption bands that appeared at 1850 cm^{-1} and 1905 cm^{-1} in Ni/TiO₂-OH and Ni/TiO₂-Ref1, respectively, could be attributed to 3-fold bound carbonyls on Ni sites (Figure R7b). **This result implied the exposure of Ni particles after the *in-situ* RWGS reaction, which was consistent with the *ex-situ* EELS-mapping results that Ni particles on TiO₂-Ref1 support were not completely covered by TiO₂.** In order not to mislead the readers, we emphasized on page7, line 11 of the revised manuscript that the EELS-mapping was obtained by ex situ.

Figure R7. *In situ* diffuse reflectance infrared Fourier transform (DRIFTS) spectroscopy after exposing the Ni/TiO₂ catalyst to CO at $-20\text{ }^{\circ}\text{C}$ after the RWGS reaction under $600\text{ }^{\circ}\text{C}$.

Comment 20: Figure R32: Without fits, how can the Ni-Ti be indicated? Or NiO or Ni-Ni for that matter, although in the reference materials I can live with that.

Response: Thanks for reviewer’s comment. The fitting results were shown in Table R2. This table was also shown as Supplementary Table 1.

Table R2. EXAFS fitting results of the fresh and used 10Ni/TiO₂-OH catalysts.

Catalysts		$\Delta E_0(\text{eV})$
10Ni/TiO ₂ -OH	Ni–O	-2.947 ± 0.980

fresh				
	R(Å)	CN	σ^2 (Å ²)	
	2.021 ± 0.009	3.316 ± 0.423	0.003 ± 0.001	
				
	Ni–Ti			
				
	R(Å)	CN	σ^2 (Å ²)	
	3.017 ± 0.013	2.692 ± 0.602	0.007 ± 0.002	
				
	Ni–Ni			
10Ni/TiO ₂ -OH				
	R(Å)	CN	σ^2 (Å ²)	-5.660 ± 1.120
used				
	2.485 ± 0.006	7.941 ± 0.737	0.005 ± 0.001	

Comment 21: “uncovered by TiO₂ under RWGS reaction conditions,” >>> change to not covered, uncovered here is a bit ambiguous.

Response: We appreciate the reviewer’s comment. We have modified the description of “uncovered” to “not covered”.

Comment 22: “CO molecules could adsorb on Ni⁰ located at 2093 cm⁻¹ and Ni²⁺ located at 2152 cm⁻¹ (Figure R39). The Figure R38a has been shown in revised manuscript as Figure 5a.” >>> The peak assignment in Figure 5C should be further discussed. 2090 cm⁻¹ is typically already sub-carbonyl species on Ni, Ni tetra carbonyl can even form. 2060 is more in line with Ni⁰ on NPs.

Response: We appreciate the reviewer’s valuable comments. As the reported literatures (Jensen, M. B. et al. *Catal. Today* **2012**, 197, 38–49; Zarfl, J. et al. *Appl. Catal. A: Gen.* **2015**, 495, 104–114), the signal at 2090 cm⁻¹ likely corresponded to sub-carbonyl Ni(CO)_x (x = 2 or 3) species, which featured a highly disordered structure and were attributed to amorphous Ni sites. Moreover, the sub-carbonyl species were weakly adsorbed, which was favored by CO removal. In addition, physisorbed Ni(CO)₄ was located at 2045 cm⁻¹–2055 cm⁻¹. **The further discussion has been added to the revised manuscript on lines 6–12, page 15 (highlight in yellow).**

Comment 23: Fig R39: 10Ni/TiO₂-OH catalysts to CO with different partial pressure at 130 K after RWGS reaction. >>> how meaningful are these tests/the results? Clearly, the active catalyst is expected to have a relatively low CO adsorption strength. CO adsorption measurements where the pressure is different to that of the reaction conditions thus are not really meaningful; please discuss the results in light of that statement and also the previous comment. Also, the way the figure is ordered is strange, why not from low to high pressure, instead of from low to high to lower? And why not list CO in the top figure as in the bottom 2?

Response: Thanks for the reviewer's valuable comment.

(1) The *in-situ* IR experiment of CO adsorption at 130 K was not designed to test the CO adsorption strength. At the high temperature under the RWGS reaction and at room temperature after the RWGS reaction, the adsorption capacity of CO on the catalyst was very weak. For this reason, **it was difficult to use CO molecules as probes to characterize the electronic structure of the Ni species under and after the RWGS reaction by *in-situ* DRIFTS experiments at high temperature and room temperature.** In general, the decrease of temperature could improve the adsorption strength of metal species for CO molecules, so as to more clearly characterize the electronic structure of metal species after the reaction (*Shen, W. J. et al. Nat. Catal. 2019, 2, 334–341*). Therefore, *in situ* IR spectra of CO adsorption on 10Ni/TiO₂-OH catalysts were collected at a low testing temperature of 130 K after the H₂ pretreatment and RWGS reaction. Although the injected CO pressure was low, it could still be used as probe molecules to identify the different Ni species. Meanwhile, the temperature of the *in situ* cell was also able to remain constant due to the low amount of CO. The obtained results demonstrated that the dominant Ni²⁺-CO signal confirmed the presence of rich Ni²⁺, which presented the strong EMSI between Ni and TiO₂. In addition, sub-carbonyl Ni(CO)_x (x = 2 or 3) species were weakly adsorbed, which was favored by CO removal, thereby facilitating the high CO selectivity of the RWGS reaction. **We have added the more detailed discussion to the revised manuscript on page 15.**

(2) The order of the figures was based on the order of CO injection pressure during the experiment. After the samples were treated with H₂ or RWGS at 873 K, the temperature

was reduced to 130K. Subsequently, CO was injected in turn at the pressure of 1×10^{-3} mbar and 1×10^{-2} mbar accompanied by data collection. Then the sample cell was re-evacuated to 1×10^{-7} mbar and data were collected again. At this time, CO almost did not exist in the sample cell, so CO was not listed. The *in-situ* IR results at 130 K under the pressure of 1×10^{-2} mbar with CO and 1×10^{-7} mbar without CO jointly confirmed the electronic structure after H₂ pretreatment and RWGS reaction. **To more clearly compare the electronic structures of the samples, the signals at same pressures were compared, which were shown in the revised manuscript as Figure 5c and Figure R8b has shown in the revised SI as Supplementary Fig. 31.**

Comment 24: "The former signal related to the CO adsorption on Ti⁴⁺ sites was also found on pure TiO₂ 52 (Supplementary Fig. 30b)." >>> what about IR of the TiO₂-OH sample? I find the 2152 to be dubious and it must be ruled out that it is not related to the TiO₂.

Response: Thanks for reviewer's valuable comments. *In-situ* infrared spectra for pristine TiO₂-OH sample after H₂ pretreatment and 10Ni/TiO₂-OH catalysts after H₂ pretreatment and RWGS reaction were shown as Figure R8. **Pristine TiO₂-OH sample exhibited a main peak at 2168 cm⁻¹ with a shoulder peak at 2147 cm⁻¹, ascribed to the adsorption of CO on distinct exposed facets of TiO₂.** By contrast, **there were two extra peaks in Ni/TiO₂-OH catalyst. One at 2152 cm⁻¹ was assigned to CO linearly adsorbed on Ni²⁺ sites.** In addition, the signal at 2090 cm⁻¹ likely corresponded to sub-carbonyl Ni(CO)_x (x = 2 or 3) species, which featured a highly disordered structure and were attributed to amorphous Ni sites (Figure R8a). Furthermore, similar results were observed as the CO partial pressure was reduced to 1×10^{-7} mbar (Figure R8b). **Figure R8a has been shown in the revised manuscript as Figure 5c and Figure R8b has shown in the revised SI as Supplementary Fig. 31. The detailed discussions were given on lines 6–12, Page 15 (highlight in yellow).**

Figure R8 *In-situ* infrared spectra recorded after exposing the $\text{TiO}_2\text{-OH}$ after H_2 pretreatment and $10\text{Ni}/\text{TiO}_2\text{-OH}$ catalysts after H_2 pretreatment and RWGS reaction to CO at 130 K with (a) 1×10^{-2} mbar and (b) 1×10^{-7} mbar.

Comment 25: *10Ni/TiO₂-OH remained stable with the CO₂ conversion more than ~55% and showed almost complete CO selectivity. In contrast, under relative milder reaction conditions, 10Ni/TiO₂-Ref1 lost almost all of its activity in only 2.5 hours (Supplementary Fig. 26a). >>> why are the reference and main cat not compared under the same long-term stability conditions?*

Response: Thanks for reviewer's valuable comments and suggestions. In this manuscript, we used a GHSV of $400,000 \text{ mL} \cdot \text{g}_{\text{cat}}^{-1} \cdot \text{h}^{-1}$ to evaluate the temperature-dependent activity. However, the CO_2 conversion of $10\text{Ni}/\text{TiO}_2\text{-OH}$ reached the thermodynamic limitation at $600 \text{ }^\circ\text{C}$. Under such a GHSV, the stability of the catalyst could not be scientifically accessed. **In order to better track the deactivation behavior, we used a much higher GHSV ($800,000 \text{ mL} \cdot \text{g}_{\text{cat}}^{-1} \cdot \text{h}^{-1}$) in the long-term stability test.** By contrast, the $10\text{Ni}/\text{TiO}_2\text{-Ref1}$ catalyst exhibited the low CO_2 conversion less than 10% under a GHSV of $400,000 \text{ mL} \cdot \text{g}_{\text{cat}}^{-1} \cdot \text{h}^{-1}$ even at $600 \text{ }^\circ\text{C}$. Therefore, it was meaningless to increase the GHSV to $800,000 \text{ mL} \cdot \text{g}_{\text{cat}}^{-1} \cdot \text{h}^{-1}$ to examine stability of $10\text{Ni}/\text{TiO}_2\text{-Ref1}$ catalyst. Thanks for reviewer's comments again.

Comment 26: “The strong interaction between intrinsic OH groups was proven”.

Response: We appreciate the comments raised by the referee. We have rephrased the “isolated Ni atoms” to “highly dispersed Ni species” on line 29, page 18.

Comment 27: References missing in main text: F. X. Xiao et al. *ACS Catal.* 2023, 13, 10500–10510; Vogt, C. et al. *Nat. Comm.* 2019, 10, 5330; J. Wang, et al. *ACS Catal.* 2021, 11, 6081–6090; Bao, X. H. et al. *J. Am. Chem. Soc.* 2022, 144, 11, 4874–4882.

Response: Thank the reviewer’s valuable for suggestion. Xiao et al. reported an efficient strategy for boosting H₂ production through the SMSI-mediated interfacial hydroxyls at low temperatures, which suggested an important role for hydroxyl groups. Vogt et al. have reported that via a complete theoretical mechanistic understanding of CO₂ hydrogenation over a classical monometallic catalyst system, Ni on silica, they are entering the era of rational design and are able to tune the selectivity of CO₂ conversion experimentally. The excellent work deepened our understanding of selective CO₂ hydrogenation reactions. Wang et al. reported that Pt nanoparticles were encapsulated by an amorphous and permeable TiO_x overlayer driven by melamine in an oxidative condition, which revealed the properties of TiO₂-coated nanoparticles. Bao et al. reported the SMSI state can be constructed in a Ru-MoO₃ catalyst using CO₂ hydrogenation reaction gas and at a low temperature of 250 °C, which favors the selective CO₂ hydrogenation to CO. This work suggested a strategy to obtain highly selective CO products through SMSI. We consider the above references to be very valuable and have already cited them as **references 35, 9, 40 and 16** (highlighted in yellow) in revised manuscript, respectively.

REVIEWERS' COMMENTS

Reviewer #3 (Remarks to the Author):

The authors have carefully addressed my concerns. The manuscript is acceptable for publication in Nature Communications as it is.

Reviewer #4 (Remarks to the Author):

The authors did an excellent job addressing the concerns. I support the publication of the article in Nature Communications.